# Sub-Sequential Physics-Informed Learning with State Space Model

**Chenhui Xu** [1]  **Dancheng Liu** [1]  **Yuting Hu** [1]  **Jiajie Li** [1]  **Ruiyang Qin** [1 2]  **Qingxiao Zheng** [1]  **Jinjun Xiong** [1]

## Abstract

Physics-Informed Neural Networks (PINNs) are a kind of deep-learning-based numerical solvers for partial differential equations (PDEs). Existing PINNs often suffer from failure modes of being unable to propagate patterns of initial conditions. We discover that these failure modes are caused by the simplicity bias of neural networks and the mismatch between PDE's continuity and PINN's discrete sampling. We reveal that the State Space Model (SSM) can be a continuous-discrete articulation allowing initial condition propagation, and that simplicity bias can be eliminated by aligning a sequence of moderate granularity. Accordingly, we propose PINN-Mamba, a novel framework that introduces sub-sequence modeling with SSM. Experimental results show that PINNMamba can reduce errors by up to 86.3% compared with state-of-the-art architecture. Our code is available at https://github.com/miniHuiHui/PINNMamba.

## 1. Introduction

In the past few years, Physics-Informed Neural Networks (PINNs) (Raissi et al., 2019) have emerged as a novel approach for numerically solving partial differential equations (PDEs). PINN takes a neural network $u_\theta(x, t)$, whose parameters $\theta$ are trained with physics PDE residual loss, as the numerical solution $u(x, t)$ of the PDE, where $x$ and $t$ are spatial and temporal coordinates. The core idea behind PINNs is to take advantage of the universal approximation property of neural networks (Hornik et al., 1989) and automatic differentiation implemented by mainstream deep learning frameworks, such as PyTorch (Paszke et al., 2019) and Tensorflow (Abadi et al., 2016), so that PINNs can achieve potentially more precise and efficient PDE solution approximation compared with traditional numerical



|  |  |  |
|---|---|---|
| (a) Ground Truth | (b) PINN | (c) PINNMamba |

*Figure 1.* PINN gradually distorts on convection equation.

approaches like finite element methods (Reddy, 1993).

The mainstream PINNs predominantly employ multilayer perceptrons (MLPs) as their backbone architecture. However, despite the universal approximation capability of MLPs, they do not always guarantee the accurate learning of numerical solutions to PDEs in practice. This phenomenon is observed as the failure modes in PINNs, in which case PINN provides a completely wrong approximation (Krishnapriyan et al., 2021). As illustrated in Fig. 1, the failure modes often manifest as a temporal gradual distortion. This distortion arises because MLPs lack the necessary inductive bias to effectively capture the temporal dependencies of a system, ultimately hindering the accurate propagation of physical patterns informed by the initial conditions.

To introduce such an inductive bias, several sequence-to-sequence approaches have been proposed (Krishnapriyan et al., 2021; Zhao et al., 2024; Yang et al., 2022; Gao et al., 2022). Specifically, Krishnapriyan et al. (2021) propose training a new network at each time step and recursively using its output as the initial condition for the next step. This method incurs significant computational and memory overhead while exhibiting poor generalization. Furthermore, Transformer-based approaches (Zhao et al., 2024; Yang et al., 2022; Gao et al., 2022) are proposed to address the time-dependency issue. Yet, these methods are based on discrete sequences, making their model produce incorrect pattern mutations in some cases due to their ignorance of the basic principle that PINNs approximate continuous dynamical systems. Thus, there is still an open question:

*How can we effectively introduce sequentiality to PINNs?*

To answer this question, we need to understand the essential difficulties of training PINNs. First, PINNs assume temporal continuity, whereas, during their actual training, the spatio-temporal collocation points used to construct the PDE residual loss are sampled discretely. We define this nature of

---

[1]University at Buffalo, SUNY [2]University of Notre Dame. Correspondence to: Chenhui Xu <cxu26@buffalo.edu>, Jinjun Xiong <jinjun@buffalo.edu>.

*Proceedings of the $42^{nd}$ International Conference on Machine Learning*, Vancouver, Canada. PMLR 267, 2025. Copyright 2025 by the author(s).

PINN as *Continuous-Discrete Mismatch*. In the absence of a well-defined continuous-discrete articulation, the real trajectory of physical system is not necessarily recovered correctly in the training process, since such *Continuous-Discrete Mismatch* would block the propagation of the initial condition.

To respect the inherent *Continuous-Discrete Mismatch*, we reveal that the State Space Models (SSM) (Kalman, 1960) can be a good continuous-discrete articulation. SSMs parametrically model a discrete temporal sequence as a continuous dynamic system. An SSM's discrete form approximates the instantaneous states and rates of change of its continuous form via integrating the system's dynamics over each time interval, which more accurately responds to the trajectory of a continuous system. Meanwhile, the SSM unifies the scale of derivatives of different orders, making it easier to be optimized. So far, SSMs have shown their insane capacity in language (Gu & Dao, 2023) and vision (Liu et al., 2024a) tasks, but its potential for solving PDEs remains unexplored. We propose to construct PINNs with SSMs to unleash their excellent properties of continuous-discrete articulation.

Next, we remark that the simplicity bias (Shah et al., 2020) of neural networks is another crucial contributing factor to PINN training difficulty. The simplicity bias will lead the model to choose the pattern with the simplest hypothesis. This results in an over-smoothed solution when approximating PDEs. Because, for data-free PINNs, there might be a very simple function in the feasible domain that can make the residual loss zero. For example, for convection equation, $\bar{u}(x, t) = 0$ leads to zero empirical loss on every collocation point except when $t = 0$. While the correct pattern is hard to fight against over-smoothed patterns during training.

A major way to eliminate simplicity bias is to construct agreements over diversity predictions (Teney et al., 2022). Following this principle, we propose a novel sub-sequence alignment approach, which allows the diverse predictions of time-varying SSMs to form such agreements. Sub-sequence modeling adopts a medium sequence granularity, forming the time dependency that a small sequence fails to capture while avoiding the optimization problem along with the long sequence. Meanwhile, the alignment of the sub-sequence predictions ensures the global pattern propagation and the formation of an agreement that eliminates simplicity bias.

In this paper, we introduce a novel learning framework to solve physics PDE's numerically, named PINNMamba, which performs time sub-sequences modeling with the Selective SSMs (Mamba) (Gu & Dao, 2023). PINNMamba successfully captures the temporal dependence within the PDE when training the continuous dynamical systems with discretized collocation points. To the best of our knowledge, PINNMamba is the first data-free SSM-based model that effectively solve physics PDE. Experiments show that PINN-Mamba outperforms other PINN approaches such as PINNs-

Former (Zhao et al., 2024) and KAN (Liu et al., 2024c) on multiple hard problems, achieving a new state-of-the-art.

**Contributions.** We make the following contributions:

- We reveal that the mismatch between the discrete nature of the training collocation points and the continuous nature of the function approximated by the PINNs is an important factor that prevents the propagation of the initial condition pattern over time in PINNs.
- We also note that the simplicity bias of neural networks is a key contributing factor to the over-smoothing pattern that causes gradual distortion in PINNs.
- We propose PINNMamba, which eliminates the discrete-continuity mismatch with SSM and combats simplicity bias with sub-sequential modeling, resulting in state-of-the-art on several PINN benchmarks.

## 2. Related Works

**Physics-Informed Neural Networks.** Physics-Informed Neural Networks (Raissi et al., 2019) are a class of deep learning models designed to solve problems governed by physical laws described in PDEs. They integrate physics-based constraints directly into the training process in the loss function, allowing them to numerically solve many key physical equations, such as Navier-Stokes equations(Jin et al., 2021), Euler equations (Mao et al., 2020), heat equatuons (Cai et al., 2021). Several advanced learning schemes such as gPINN (Kharazmi et al., 2019), vPINN(Yu et al., 2022), and RoPINN(Wu et al., 2024), model architectures such as QRes (Bu & Karpatne, 2021), FLS (Wong et al., 2022), PINNsFormer (Zhao et al., 2024), KAN (Liu et al., 2024c;b) are proposed in terms of convergence, optimization, and generalization.

**Failure Modes in PINNs.** Despite these efforts, PINN still has some inherently intractable failure modes. Krishnapriyan et al. (2021) identify several types of equations that are vulnerable to difficulties in solving by PINNs. These equations are usually manifested by the presence of a parameter in them that makes their pattern behave as a high frequency or a complex state (Cho et al., 2024), failing to propagate the initial condition. In such cases, an empirical loss constructed using a collocation point can easily fall into an over-smooth solution (e.g. $\bar{u}(x, t) = 0$ can make the loss of all collocation points except whose $t = 0$ descend to $0$ for 1d-wave equations). Several methods regarding optimization (Wu et al., 2024; Wang et al., 2022a), sampling (Gao et al., 2023; Wu et al., 2023), model architecture (Zhao et al., 2024; Cho et al., 2024; Nguyen et al., 2024b), transfer learning (Xu et al., 2023; Cho et al., 2024) are proposed to mitigate such failure modes. However, the above approaches do not focus on the fact that a PDE system should be modeled as a continuous dynamic, leading to difficulties in generalization over a wide range of problems.

**State Space Models.** The state space model (Kalman, 1960) is a mathematical representation of a physical system in terms of state variables. Modern SSMs (Gu et al., 2022; Smith et al., 2023; Gu & Dao, 2023) combine the representational power of neural networks with their own superior long-range dependency capturing and parallel computing capabilities and thus are widely used in many fields, such as language modeling (Fu et al., 2023; Poli et al., 2023; Gu & Dao, 2023; Dao & Gu, 2024), computer vision (Zhu et al., 2024; Liu et al., 2024a), and genomics (Gu & Dao, 2023; Nguyen et al., 2024a). Specifically, Structured SSMs (S4) (Gu et al., 2022) decomposing the structured state matrices as the sum of a low-rank and normal terms to improve the efficiency of state-space-based deep models. Further, Selective SSMs (Mamba) (Gu & Dao, 2023) eliminates the Linear Time Invariance (Sain & Massey, 1969) of SSMs by introducing a gating mechanism, allowing the model to selectively propagate or forget information and greatly enhancing the model performance. In physics, SSMs are used in conjunction with Neural Operator to form a data-driven solution to PDEs (Zheng et al., 2024; Hu et al., 2024). However, these methods are data-driven which lack generalization ability in some scenarios where real data is not available. Unlike these methods, our approach, PINN-Mamba is fully physics-driven, relying only on residuals constructed using PDEs without any training data.

## 3. Preliminary

**Physics-Informed Neural Networks.** The PDE systems that are defined on spatio-temporal set $\Omega \times [0, T] \subseteq \mathbb{R}^{d+1}$ and described by equation constraints, boundary conditions, and initial conditions can be formulated as:

$$\mathcal{F}(u(x,t)) = 0, \forall (x,t) \in \Omega \times [0,T]; \quad (1)$$

$$\mathcal{I}(u(x,t)) = 0, \forall (x,t) \in \Omega \times \{0\}; \quad (2)$$

$$\mathcal{B}(u(x,t)) = 0, \forall (x,t) \in \partial\Omega \times [0,T], \quad (3)$$

where $u : \mathbb{R}^{d+1} \to \mathbb{R}^m$ is the solution of the PDE, $x \in \Omega$ is the spatial coordinate, $\partial\Omega$ is the boundary of $\Omega$, $t \in [0, T]$ is the temporal coordinate and $T$ is the time horizon. The $\mathcal{F}, \mathcal{I}, \mathcal{B}$ denote the operators defined by PDE equations, initial conditions, and boundary conditions respectively.

A physics-driven PINN first builds a finite collocation point set $\chi \subset \Omega \times [0, T]$, and its spatio (temporal) boundary $\partial\chi \subset \partial\Omega \times [0, T]$ ($\chi_0 \subset \Omega \times \{0\}$), then employs a neural network $u_\theta(x, t)$ which is parameterized by $\theta$ to approximate $u(x, t)$ by optimizing the residual loss as defined in Eq. 7:

$$\mathcal{L}_{\mathcal{F}}(u_\theta) = \frac{1}{|\chi|} \sum_{(x_i,t_i) \in \chi} \|\mathcal{F}(u_\theta(x_i, t_i)\|^2; \quad (4)$$

$$\mathcal{L}_{\mathcal{I}}(u_\theta) = \frac{1}{|\chi_0|} \sum_{(x_i,t_i) \in \chi_0} \|\mathcal{I}(u_\theta(x_i, t_i)\|^2; \quad (5)$$



*Figure 2.* Failure mode of PINN on convection equation, the oversmooth solution brings the losses down to 0 almost everywhere.

$$\mathcal{L}_{\mathcal{B}}(u_\theta) = \frac{1}{|\partial\chi|} \sum_{(x_i,t_i) \in \partial\chi} \|\mathcal{B}(u_\theta(x_i, t_i)\|^2; \quad (6)$$

$$\mathcal{L}(u_\theta) = \lambda_{\mathcal{F}}\mathcal{L}_{\mathcal{F}}(u_\theta) + \lambda_{\mathcal{I}}\mathcal{L}_{\mathcal{I}}(u_\theta) + \lambda_{\mathcal{B}}\mathcal{L}_{\mathcal{B}}(u_\theta), \quad (7)$$

where $\lambda_{\mathcal{F}}, \lambda_{\mathcal{I}}, \lambda_{\mathcal{B}}$ are the weights for loss that are adjustable by auto-balancing or hyperparameters. $\|\cdot\|$ denotes $l^2$-norm.

**State Space Models.** An SSM describes and analyzes a continuous dynamic system. It is typically described by:

$$\dot{\mathbf{h}}(\mathbf{t}) = A\mathbf{h}(t) + B\mathbf{x}(t), \quad (8)$$

$$\mathbf{u}(t) = C\mathbf{h}(t) + D\mathbf{x}(t), \quad (9)$$

where $\mathbf{h}(t)$ is hidden state of time $t$, $\dot{\mathbf{h}}(t)$ is the derivative of $\mathbf{h}(t)$. $\mathbf{x}(t)$ is the input state of time $t$, $\mathbf{u}(t)$ is the output state, and $A, B, C, D$ are state transition matrices.

In real-world applications, we can only sample in discrete time for building a deep SSM model. We usually omit the term $D\mathbf{x}(t)$ in deep SSM models because it can be easily implemented by residual connection (He et al., 2016). So we create a discrete time counterpart:

$$\mathbf{h}_k = \bar{A}\mathbf{h}_{k-1} + \bar{B}\mathbf{x}_k, \quad (10)$$

$$\mathbf{u}_k = C\mathbf{h}_k, \quad (11)$$

with discretization rules such as zero-order hold (ZOH):

$$\bar{A} = \exp(\Delta A), \quad (12)$$

$$\bar{B} = (\Delta A)^{-1}(\exp(\Delta A) - I) \cdot \Delta B, \quad (13)$$

where $\bar{A}$ and $\bar{B}$ is discrete time state transfer and input matrix, and $\Delta$ is a step size parameter. By parameterizing $A$ using HiPPO matrix (Gu et al., 2020), and parameterizing $(\Delta, B, C)$ with input-dependency, a time-varying Selective SSM can be constructed (Gu & Dao, 2023). Such a Selective SSM can capture the long-time continual dependencies in dynamic systems. We will argue that this makes SSM a good continuous-discrete articulation for modeling PINN.

## 4. Why PINNs Present Failure Modes?

A counterintuitive fact of PINNs is that the failure modes are not devoid of optimizing their residual loss to a very low

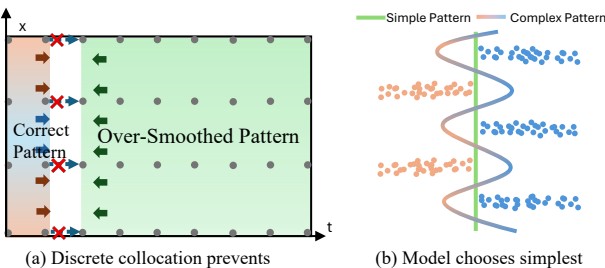

(a) Discrete collocation prevents the correct pattern to propagate.

(b) Model chooses simplest pattern due to Simplicity Bias.

*Figure 3.* The correct Pattern determined by the initial conditions faces two resistances in propagation: (a) the difficulty of propagating information directly through the gradient among discrete collocation points, and (b) the need to fight against over-smoothed solutions with near-zero loss caused by simplicity bias.

degree. As shown in Fig. 2, for the convection equation, the converged PINN almost completely crashes in the domain, but its loss maintains a near-zero degree at almost every collocation point. This is the result of the combined effects of the simplicity bias (Shah et al., 2020; Pezeshki et al., 2021) of neural networks and the *Continuous-Discrete Mismatch* of PINNs, as shown in Fig. 3. The simplicity bias is the phenomenon that the model tends to choose the one with the simplest hypothesis among all feasible solutions, which we demonstrate in Fig. 3 (b). *Continuous-Discrete Mismatch* refers to the inconsistency between the continuity of the PDE and the discretization of PINN's training process. As shown in Eq. 4 - 6, to construct the empirical loss for the PINN training process, we need to determine a discrete and finite set of collocation points on $\Omega \times [0, T]$. This is usually done with a grid or uniform sampling. But a PDE system is usually continuous and its solutions should be regular enough to satisfy the differential operator $\mathcal{F}$, $\mathcal{B}$, and $\mathcal{I}$.

**Continuous-Discrete Mismatch.** *Continuous-Discrete Mismatch* will cause correct local patterns hard to propagate over the global domain. Because the loss on discrete collocation points does not necessarily respond to the correct pattern on the continuous domain, instead, only responds to its small neighborhood. To show such *Continuous-Discrete Mismatch*, we first present the following theorem:

**Theorem 4.1.** *Let* $\chi^* = \{(x_1^*, t_1^*), \ldots, (x_N^*, t_N^*)\} \subset \Omega \times [0, T]$. *Then for differential operator* $\mathcal{M}$ *there exist infinitely many functions* $u_\theta : \Omega \to \mathbb{R}^m$ *parametrized by* $\theta$, *s.t.*

$$\mathcal{M}(u_\theta(x_i^*, t_i^*)) = 0 \quad for\ i = 1, \ldots, N,$$

$$\mathcal{M}(u_\theta(x, t)) \neq 0 \quad for\ a.e.\ x \in \Omega \times [0, T] \setminus \chi^*.$$

By Theorem 4.1, enforcing the PDE only at a finite set of points does not guarantee a globally correct solution. This can be performed by simply constructing a Bump function in a small neighborhood of points in $\chi^*$ so that it satisfies $\mathcal{M}(u_\theta(x^*, t^*)) = 0$ for $(x^*, t^*) \in \chi^*$. This means that

the information of the equation determined by the initial conditional differential operator $\mathcal{I}$ may act only on a small neighborhood of collocation points with $t = 0$. The other collocation points in the $\Omega \times (0, T]$, on the other hand, might fall into a local optimum that can make $\mathcal{L}_\mathcal{F}(u_\theta)$ defined by Eq. 4 to near 0. Because the function $u_\theta$ determined by $\mathcal{F}$ and $\mathcal{I}$ together on the collocation points at $t = 0$ may not be generalized outside its small neighborhood. The detailed proof of Theorem 4.1 can be found in Appendix A.

**Simplicity Bias.** Meanwhile, the simplicity bias of neural networks will make the PINNs always tend to choose the simplest solution in optimizing $\mathcal{L}_\mathcal{F}(u_\theta)$. This implies that PINN will easily fall into an over-smoothed solution. For example, as shown in Fig. 2, the PINN's prediction is 0 in most regions. The loss of this over-smoothed feasible solution is almost identical to that of the true solution, and the existence of an insurmountable ridge between the two loss basins results in a PINN that is extremely susceptible to falling into local optimums. As in Fig 3, the over-smoothed pattern yields an advantage against the correct pattern.

Under the effect of difficulty in passing locally correct patterns to the global due to *Continuous-Discrete Mismatch* and over-smoothing due to simplicity bias, PINNs present failure modes. Therefore, to address such failure modes, the key points in designing the PINN models lie in: (1) a mechanism for information propagation in continuous time and (2) a mechanism to eliminate the simplicity bias of models.

## 5. Combating Failure Mode with State-Space Model and Sub-sequential Alignment

To address the problems in Section 4, we propose (1) a discrete state-space-based encoder that models the sequences of individual collocation points in continuous dynamics, to match with *Continuous-Discrete Mismatch*, and propagates the information from the initial condition to subsequent times (Section 5.1). and (2) a sub-sequence contrastive alignment mechanism that aligns different outputs of the same collocation point in different sub-sequences, to form an agreement that eliminates simplicity bias (Section 5.2).

### 5.1. Continuous Time Propagation of Initial Condition Information with State Space Model

As we discussed in Section 4, the *Continuous-Discrete Mismatch* of PINNs raises the intrinsic difficulty of modeling, since the time dependency in a dynamic PDE system is not captured spontaneously by discrete sampling. We argue that such a dynamic time dependency can be modeled by SSM. To this end, we first consider the PDE as a spatially infinite-dimensional ODE to simplify the problem. We view the solution $u_\theta(x, t)$ in a function space that, if we let:

$$U(t) := u_\theta(\cdot, t), \tag{14}$$

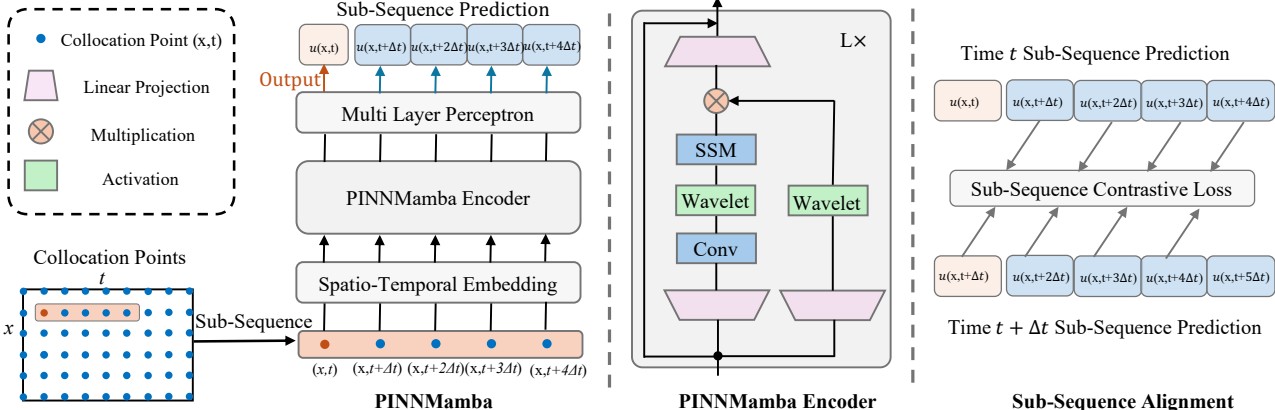

*Figure 4.* PINNMamba Overview. PINNMamba takes the sub-sequence as input which is a composite of several consecutive collocation points on the time axis. For each sub-sequence, the prediction of the first collocation point is taken as the output of PINNMamba, while the others are used to align the prediction of different sub-sequences, that can propagate information among time coordinates.

be a function $x \rightarrow u_\theta(x, t)$, by $M$-point spatial sampling:

$$U_i(t) := u(x_i, t), \tag{15}$$

$$\mathbf{u}(t) = [U_1(t), U_2(t), \cdots, U_M(t)]^\top. \tag{16}$$

**Sequential Modeling Continuity with SSM.** In continuous time, we now model the function $\mathbf{u}(t)$ to the dynamic system described by SSM as in Eq. 8 and 9. Here we let $\mathbf{x}(t) = \text{Embed}(x, t)$, where $\text{Embed}(\cdot)$ is the Spatio-Temporal Embedding in Fig 4. After temporal discretization $\mathbf{u}_k = \mathbf{u}(k\Delta t), \mathbf{h}_k = \mathbf{h}(k\Delta t)$ and $\mathbf{x}_k = \mathbf{x}(k\Delta t)$, we get:

$$\mathbf{u}_k = C\bar{A}^k \mathbf{h}_0 + C \sum_{i=0}^{k} \bar{A}^{k-i} \bar{B} \mathbf{x}_i. \tag{17}$$

Reversibly, by the inverse of the discretization rule defined by Eq. 12, 13, we can restore this temporal dependency to continuous time. This kind of restoration can help achieve PINN's generalization to any moment in $[0, T]$.

**Pattern Propagation by Joint Optimization.** Combine Eq. 4 with 17, in a sequence start with $t = 0$, the sum of loss of collocation points at time $k\Delta t$, would be:

$$\sum_{i=1}^{M} \mathcal{L}_\mathcal{F}(u(x_i, k\Delta t)) = \frac{1}{M} \|\mathcal{F}(\mathbf{1}_M \cdot \mathbf{u}_k)\|^2$$

$$= \frac{1}{M} \left\| \mathcal{F}\left(\mathbf{1}_M \cdot (C\bar{A}^k \mathbf{h}_0 + C \sum_{i=0}^{k} \bar{A}^{k-i} \bar{B} \mathbf{x}_i)\right) \right\|^2, \tag{18}$$

where $\mathbf{1}_M = [1, 1, \cdots, 1] \in \mathbb{R}^M$. In Eq. 18, we notice that the $\mathbf{h}_0$ should satisfy both the initial condition and the equation by jointly optimizing the losses:

$$\mathcal{L}_\mathcal{F}(\mathbf{u}_0) = \frac{1}{M} \|\mathcal{F}(\mathbf{1}_M \cdot (C\mathbf{h}_0))\|^2; \tag{19}$$

$$\mathcal{L}_\mathcal{I}(\mathbf{u}_0) = \frac{1}{M} \|\mathcal{I}(\mathbf{1}_M \cdot (C\mathbf{h}_0))\|^2. \tag{20}$$

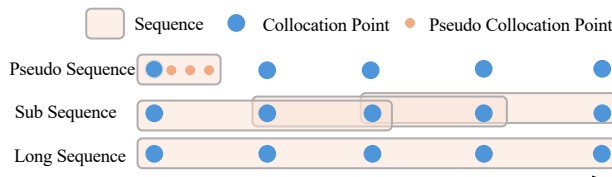

*Figure 5.* Comparison of Sequence Granularity

Thereby, for each collocation point, the numerical value of its solution should be jointly optimized by Eq. 18, 19, and 20, thus receiving the pattern defined by the initial conditions.

**Uniformed Derivatives Scale.** Another benefit that can be got from SSM is, by parameterizing differential state matrix $A$ in Eq.8 with HiPPO matrix (Gu et al., 2020) which contains the derivative information, we can align the derivatives of the system with respect to time on a uniform scale. This uniform scale will help to reduce the problem of ruggedness on the loss landscape due to gradient vanishing or exploding.

**Time-Varying SSM.** In practice, we use the time-varying Selective SSM (Gu & Dao, 2023), instead of the function defined by Eq. 17 being the SSM on a linear time-invariant system. The time-varying SSM has two advantages, one is that such input-dependent models typically have stronger representational capabilities (Xu et al., 2024), while the other is that it will make diverse predictions that help to eliminate simplicity bias in the model, as we will discuss in section 5.2. This time-variance will make $(\bar{A}, \bar{B}, C)$ time-dependent, and therefore, Eq. 17 and 18 need minor adjustments. These adjustments won't impact the initial condition propagation, and we will discuss them in Appendix B.

### 5.2. Eliminating Simplicity Bias of Models with Sub-Sequence Contrastive Alignment

Although SSM can make the information about the initial conditions propagate in time coordinates, it still cannot mit-

igate the simplicity bias of neural networks. The model is still prone to falling into an over-smoothed local optimum. There are two key points to address this over-smoothness caused by simplicity bias: (1) appropriate sequence granularity to guarantee a smooth optimization process. (2) Mitigating the effect of simplicity bias through the diversity of model prediction paradigms (Pagliardini et al., 2023).

**Sequence Granularity.** A proper sequence granularity ensures smooth propagation of the initial conditions while making the model easier to optimize. As shown in Fig. 5, there are three ways to define sequence, which are pseudo sequence (Zhao et al., 2024), long sequence (Nguyen et al., 2024a), and the proposed sub-sequence. We propose to use a sub-sequence with medium granularity overlapping. The sub-sequential modeling can avoid: (1) the difficulty of crossing the loss barrier that makes the model trapping in the over-smooth local optimum, which is caused by the huge inertia of long sequence; (2) the difficulty of broadcasting information globally on the time coordinate, that caused by construct on small neighborhoods of a collocation point in pseudo sequence. Sub-sequence takes only the first output in the sequence as the output value of the current collocation point. Its successors' values will pass information crossing the time coordinate through subsequences alignment and form diverse predictions to eliminate simplicity bias.

**Contrastive Alignment for Information Propagation.** As shown in Fig. 5, we construct a sub-sequence for each collocation point together with its finite successors, which form overlapping collocation points. By aligning the predictions of these collocation points with a contrastive loss, each collocation point becomes a soft relay of the pattern. Thus, it forms the propagation of patterns in the whole time domain.

**Eliminating the Simplicity Bias.** Previous work (Teney et al., 2022; Pagliardini et al., 2023) has pointed out that the agreement obtained from diverse predictions is the key to eliminating the effects of simplicity bias. We argue that this agreement from diverse predictions is naturally obtained in the sub-sequence alignment. This is because the fact that, since the SSM we constructed in section 5.1 is time-varying and a collocation point will be at different time coordinates in different sub-sequences, the predictions for this collocation point are naturally diverse. And we force these diverse predictions to arrive at a consensus by contrastive alignment.

## 6. PINNMamba

In conjunction with the high-level ideas described in Section 5, in this section, we present PINNMamba, a novel physics-informed learning framework that effectively combats the failure modes in the PINNs.

**Sub-Sequential I/O.** As shown in Fig. 4, PINNMamba first samples the grid of collocation points over the entire

spatio-temporal domain bounded by the PDE. We assume that the grid picks $M$ spatial coordinates and $N$ temporal coordinates, and denote the temporal sampling interval as $\Delta t = T/(N-1)$. For a collocation point $(x, t)$, we construct a sequence $X(x, t)$ with its $k-1$ temporal successors:

$$X(x,t) = \{(x,t), (x, t+\Delta t), \cdots, (x, t+(k-1)\Delta t)\}. \quad (21)$$

PINNMamba takes such $M \times N$ sequences as the input of models. For each sequence $X(x, t)$, PINNMamba computes a sub-sequence prediction $\{\bar{u}_\theta^t(x, t), \bar{u}_\theta^t(x, t + \Delta t), \cdots, \bar{u}_\theta^t(x, t + (k - 1)\Delta t)\}$ corresponding to every collocation point in the sequence, where $\bar{u}_\theta^t(x, t + i\Delta t)$ denote the tentative prediction of collocation point $(x, t+i\Delta t)$ in a sequence start with time $t$. The $\bar{u}_\theta^t(x, t)$ will be taken as the output of collocation point $(x, t)$ and the rest of the sequence will be used to construct the sub-sequence contrastive alignment loss we will discuss later in Section 5.2. The residual losses of the model w.r.t the sub-sequence will be:

$$\mathcal{L}_\mathcal{F}^{\text{seq}}(u_\theta) = \frac{1}{k|\chi|} \sum_{(x_i,t_i)\in\chi} \sum_{j=0}^{k-1} \|\mathcal{F}(u_\theta^{t_i}(x_i, t_i + j\Delta t)\|^2; \quad (22)$$

$$\mathcal{L}_\mathcal{I}^{\text{seq}}(u_\theta) = \frac{1}{k|\chi_0|} \sum_{(x_i,t_i)\in\chi_0} \sum_{j=0}^{k-1} \|\mathcal{I}(u_\theta^{t_i}(x_i, t_i + j\Delta t)\|^2; \quad (23)$$

$$\mathcal{L}_\mathcal{B}^{\text{seq}}(u_\theta) = \frac{1}{k|\partial\chi|} \sum_{(x_i,t_i)\in\partial\chi} \sum_{j=0}^{k-1} \|\mathcal{B}(u_\theta^{t_i}(x_i, t_i + j\Delta t)\|^2. \quad (24)$$

**Model Architecture.** As shown in Fig. 4, PINNMamba employs an encoder-only architecture, which encodes fixed-size input sub-sequence into a sub-sequence prediction with the same length. First, for each token in the sequence, an MLP-based Spatio-Temporal Embedding layer first embeds the $(x, t)$ coordinates into high-dimensional representation. The embeddings will be sent to a Mamba-based encoder, which consists of several PINNMamba blocks.

The PINNMamba block employed here consists of two branches: (1) the first is a stack of a linear projection layer, a 1d-convolution layer, a Wavelet activation (Zhao et al., 2024), and an SSM layer with parallel scan (Gu & Dao, 2023); (2) the second is a stake of a linear projection layer and a Wavelet activation. The two branches are then connected with an element-wise multiplication, followed by another linear projection and residual connection. With input $X^l$, the PINNMamba block can be formulated as:

$$X_1^l = \text{SSM}(\sigma(\text{Conv}(W_a X^l))); \quad (25)$$

$$X_2^l = \sigma(W_b X^l); \quad (26)$$

$$X^{l+1} = X^l + W_c(X_1^l \otimes X_2^l), \quad (27)$$

where $\sigma(x) = \omega_1 \sin(x) + \omega_2 \cos(x)$ is Wavelet activation function (Zhao et al., 2024), in which $\omega_1, \omega_2$ are learnable. $\otimes$ denotes an element-wise multiplication.

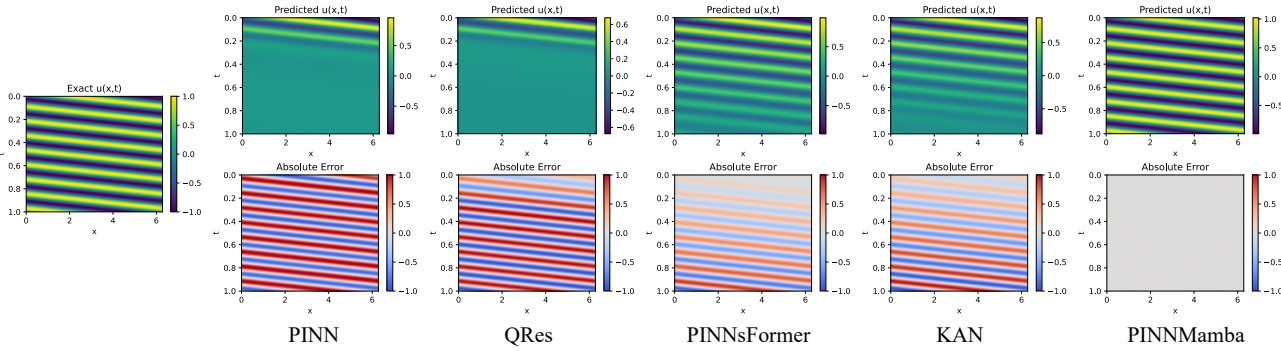

*Figure 6.* The ground truth solution, prediction (top), and absolute error (bottom) on convection equations.

**Sub-Sequence Contrastive Alignment.** PINNMamba predicts the same collocation multiple times in different subsequences. For example, the collocation point $(x, t + k\Delta t)$ appears on sequences from $X(x, t + \Delta t)$ to $X(x, t + k\Delta t)$. We align the predictions on these subsequences to make the information defined by the initial conditions propagate over time. To do this, for each subsequence, we design a contrastive loss with the last subsequence for alignment:

$$\mathcal{L}_{\text{alig}}(u_\theta) = \frac{1}{(k-1)|\chi|} \sum_{(x_i, t_i) \in \chi} \sum_{j=1}^{k-1} \left[ u_\theta^{t_i}(x_i, t_i + j\Delta t) \right.$$
$$\left. - u_\theta^{t_i + \Delta t}(x_i, t_i + j\Delta t) \right]^2. \quad (28)$$

Thus, the empirical loss for PINNMamba is defined as:

$$\mathcal{L}(u_\theta) = \lambda_{\mathcal{F}} \mathcal{L}_{\mathcal{F}}^{\text{seq}}(u_\theta) + \lambda_{\mathcal{I}} \mathcal{L}_{\mathcal{I}}^{\text{seq}}(u_\theta) + \lambda_{\mathcal{B}} \mathcal{L}_{\mathcal{B}}^{\text{seq}}(u_\theta) + \lambda_{\text{alig}} \mathcal{L}_{\text{alig}}(u_\theta). \quad (29)$$

## 7. Experiments

**Setup.** We evaluate the performance of PINNMamba on three standard PDE benchmarks: convection, wave, and reaction equations, all of which are identified as being affected by failure modes (Krishnapriyan et al., 2021; Zhao et al., 2024). The details of those PDEs can be found in Appendix C. We compare PINNMamba with four baseline models, vanilla PINN (Raissi et al., 2019), QRes (Bu & Karpatne, 2021), PINNsFormer (Zhao et al., 2024), and KAN (Liu et al., 2024c) . For fair comparison, we sample $101 \times 101$ collocation points with uniformly grid sampling, following previous work (Zhao et al., 2024; Wu et al., 2024). We also evaluate on PINNacle Benchmark (Hao et al., 2024) and Navier–Stokes equation (Raissi et al., 2019).

**Training Details.** We train PINNMamba and all the baseline models 1000 epochs with L-BFGS optimizer (Liu & Nocedal, 1989). We set the sub-sequence length to 7 for PINNMamba, and keep the original pseudo-sequence setup for PINNsFormers. The weights of loss terms $[\lambda_{\mathcal{F}}, \lambda_{\mathcal{I}}, \lambda_{\mathcal{B}}]$ are set to $[1, 1, 10]$ for all three equations, as we find that

strengthening the boundary conditions can lead to better convergence. $\lambda_{\text{alig}}$ is set to 1000 for convection and reaction equations, and auto-adapted by $\lambda_{\mathcal{F}}$ for wave equation. All experiments are implemented in PyTorch 2.1.1 and trained on an NVIDIA H100 GPU. More training details are in Appendix D. Our code and weights are available at https://github.com/miniHuiHui/PINNMamba.

**Metrics.** To evaluate the performance of the models, we take relative Mean Absolute Error (rMAE, a.k.a $\ell_1$ relative error) and relative Root Mean Square Error (rRMSE, a.k.a $\ell_2$ relative error) following common practive (Zhao et al., 2024; Wu et al., 2024). The metrics are formulated as:

$$\text{rMAE}(\hat{u}) = \frac{\sum_{n=1}^{N} |\hat{u}(x_n, t_n) - u(x_n, t_n)|}{\sum_{n=1}^{N} |u(x_n, t_n)|}, \quad (30)$$

$$\text{rRMSE}(\hat{u}) = \sqrt{\frac{\sum_{n=1}^{N} |\hat{u}(x_n, t_n) - u(x_n, t_n)|^2}{\sum_{n=1}^{N} |u(x_n, t_n)|^2}}, \quad (31)$$

where N is the number of test points, $u(x, t)$ is the ground truth solution, and $\hat{u}(x, t)$ is the model's prediction.

### 7.1. Main Results

We present the rMAE and rRMSE for approximating convection, reaction and wave equation's solution in Table 1. Our model consistently outperforms other model architectures, achieving new state-of-the-art. Notably, as shown in Fig. 6, for the convection equation, PINNMamba allows sufficient propagation of information about the initial conditions, whereas on all the other models there is a varying degree of distortion in the time coordinates. As shown in Fig. 8, PINNMamba can further optimize at the boundary, resulting in a lower error than KAN and PINNsFormer for reaction equations. For problems as intrinsically difficult to optimize as the wave, as in Fig. 7, PINNMamba effectively combats simplicity bias and aligns the scales of multi-order differentiation, and thus achieves significantly higher accuracy. This illustrates that PINNMamba can be effective against PINN's failure modes. It's also worth noting that,

*Table 1.* Results for solving convection, reaction, and wave equations.

| Model | #Params | Convection | | | Reaction | | | Wave | | |
|---|---|---|---|---|---|---|---|---|---|---|
| | | Loss | rMAE | rRMSE | Loss | rMAE | rRMSE | Loss | rMAE | rRMSE |
| PINN | 527361 | 0.0239 | 0.8514 | 0.8989 | 0.1991 | 0.9803 | 0.9785 | 0.0320 | 0.4101 | 0.4141 |
| QRes | 396545 | 0.0798 | 0.9035 | 0.9245 | 0.1991 | 0.9826 | 0.9830 | 0.0987 | 0.5349 | 0.5265 |
| PINNsFormer | 453561 | 0.0068 | 0.4527 | 0.5217 | 3e-6 | 0.0146 | 0.0296 | 0.0216 | 0.3559 | 0.3632 |
| KAN | 891 | 0.0250 | 0.6049 | 0.6587 | 7e-6 | 0.0166 | 0.0343 | 0.0067 | 0.1433 | 0.1458 |
| PINNMamba | 285763 | 0.0001 | **0.0188** | **0.0201** | 1e-6 | **0.0094** | **0.0217** | 0.0002 | **0.0197** | **0.0199** |

*Figure 7.* The ground truth solution, prediction (top), and absolute error (bottom) on wave equations.

*Table 2.* Integrating PINNMamba with advanced training strategies and loss auto-balancing strategy. The rMAE is reported here.

| Method | Convection | Reaction | Wave |
|---|---|---|---|
| PINNMamba | 0.0188 | 0.0094 | 0.0197 |
| +gPINN | 0.0172 | 0.0123 | 0.0264 |
| +vPINN | 0.0236 | 0.0092 | 0.0169 |
| +RoPINN | 0.0102 | 0.0099 | 0.0121 |
| +NTK | 0.0179 | 0.0079 | 0.0147 |
| +NTK+RoPINN | 0.0127 | 0.0072 | 0.0106 |

PINNMamba has the lowest number of parameters (except KAN), while achieving consistently the best performance.

### 7.2. Combination with Other Methods

Since PINNMamba mainly focuses on model architecture, it can be integrated with other methods effortlessly. We explore the feasibility and their performance in combination with advanced training paradigm, as well as loss balancing.

**Training Paradigm.** We show the rMAE of PINNMamba when integrated with advanced strategies in Table 2. We observe that gPINN (Yu et al., 2022) and vPINN (Kharazmi et al., 2019) erratically deliver some performance gains on some tasks. This is due to the fact that the regularization provided by gPINN and vPINN in the form of a loss function through the gradient and variational residuals has little effect on PINNMamba, since SSM itself is sufficiently regularized. RoPINN (Wu et al., 2024) reduces the PINNMamba's error on convection and wave equations by about 40%, since it complements the spatial continuity dependency.

**Neural Tangent Kernel.** Dynamic tuning of losses via Neural Tangent Kernel(NTK) (Wang et al., 2022b) has been shown to have the effect of smoothing out the loss landscape. PINNMamba also works well with the NTK-adopted loss function. As shown in Table 2, NTK can reduce PINNMamba error by 5-25%. The combination of RoPINN and NTK can further improve the overall performance of PINNMamba, which demonstrates the excellent suitability of PINNMamba with other PINN optimization methods.

### 7.3. Loss-Error Consistency Analysis

Our other interest is the role of PINNMamba for the elimination of simplicity bias. Models affected by simplicity bias that fall into over-smoothing solutions will show inconsistent decreasing trends in loss and error during training. As shown in Fig. 9, in the training process for solving convection equations, the rMAE of PINN doesn't descend as $\mathcal{L}_{\mathcal{F}}$ and $\mathcal{L}_{\mathcal{I}}$. This suggests that PINN is trapped in an over-smoothing solution, which is in agreement with our observation in Fig. 6. As a comparison, we find that PINNMamba's losses descent processes show a high degree of consistency with its error descent process. This indicates that PINNMamba does not tend to fall into a local optimum of oversimplified patterns. Instead, it tends to exhibit patterns that are consistent with the original PDEs.

### 7.4. Ablation Study

To verify the validity of the various components of the PINNMamba, as shown in Table 3, we evaluate the performance of models subtracting these components from PINNMamba.

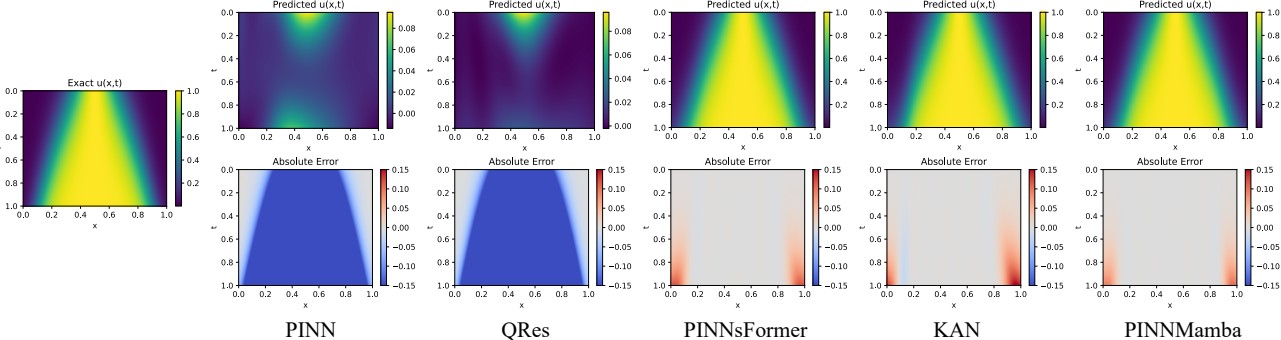

*Figure 8.* The ground truth solution, prediction (top), and absolute error (bottom) on reaction equations.

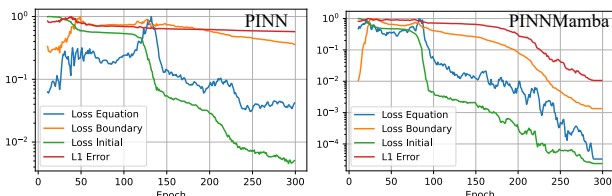

*Figure 9.* Loss and $\ell_1$-Error Curve w.r.t Training Iteration.

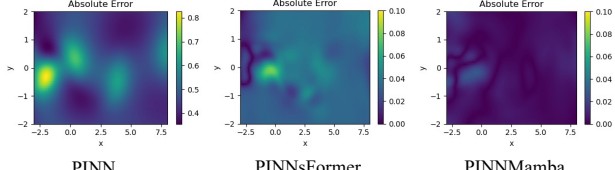

*Figure 10.* Absolute Error of pressure prediction of N-S equation

*Table 3.* Ablation Studies. The rMAE is reported here.

| Method | Convection | Reaction | Wave |
|---|---|---|---|
| PINNMamba | 0.0188 | 0.0094 | 0.0197 |
| -Sub-Sequence Align | 0.1436 | 0.0291 | 0.0481 |
| -Sub +Long Sequence | 0.6492 | 0.6731 | 0.3391 |
| -Time Varying SSM | 0.0241 | 0.0179 | 0.0664 |
| -SSM | 0.7785 | 0.9836 | 0.3341 |
| -Wavelet +Tanh | 0.4531 | 0.0299 | 0.3151 |

**Sub-Sequence.** We remove the sub-sequence alignment, which leads to a decrease in model performance, indicating the significance of the agreement formed through alignment in eliminating simplicity bias. After replacing the sub-sequence with a long sequence of the entire domain, the model shows failure modes, in line with the sequence granularity analysis in Section 5.2.

**Time-Varying SSM.** We replace the selective SSM (Gu & Dao, 2023) with a linear time-invariant structure SSM (Gu et al., 2022), and there is some decrease in model performance, illustrating the role of predictive diversity in eliminating simplicity bias. And when we remove SSM completely and switch to MLP instead, the model has severe failure modes. This demonstrates that SSM's adaptation for *Continuous-Discrete Mismatch* allows the initial condition information to propagate sufficiently in time coordinates.

In addition, we also conducted a sensitivity analysis of the choice of sub-sequence length, activation. See Appendix E.

### 7.5. Experiments on Complex Problems

To further demonstrate the generalization of our method, we tested our model on partial PINNacle Benchmark (Hao et al.,

2024) and Navier-Stokes equations. As shown in Fig. 10, PINNMamba achieves the lowest error on the N-S equation. Just like PINNsFormer, PINNMamba also gets out-of-memory on some problems in PINNacle, which we identify as a major limitation of sequence-based methods. We discuss the details of PINNacle experiments in Appendix F.

## 8. Conclusion

In this paper, we reveal that the mismatch between discrete training of PINNs and the continuous nature of PDEs, as well as simplicity bias are the key of failure modes. In combating with such failure modes, we propose PINNMamba, an SSM-based sub-sequence learning framework. PINN-Mamba successfully eliminates the failure modes, and meanwhile becomes the new state-of-the-art PINN architecture.

## Impact Statement

The development of physics-informed neural networks represents a transformative approach to solving differential equations by integrating physical laws directly into the learning process. This work explores novel advancements in PINN architecture, to improve accuracy and eliminate the potential failure modes. By refining PINN architectures, this study contributes to the broader adoption of physics-informed machine learning in fields such as computational fluid dynamics, material science, and engineering simulations. The proposed enhancements lead to more robust and scalable models, facilitating real-world applications where conventional PINNs struggle with over-smoothing. There is no known negative impact from this study at this time.

## Acknowledgements

This work is supported, in part, by the National Science Foundation and the Institute of Education Sciences under Grant 2229873 (AI4ExceptionalEd), National Science Foundation under Grant 2235364 (FuSe-TG), and SUNY-IBM AI Collaborative Research Award. Any opinions, findings and conclusions or recommendations expressed in this material are those of the author(s) and do not necessarily reflect the views of the sponsors.

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

# A. Proof of Theorem 4.1

We start with a function $v$ such that $\mathcal{M}(v)$ is non-zero almost everywhere. Such a function exists because $\mathcal{M}$ is a non-zero differential operator. For example, if $\mathcal{M}$ is the Laplacian, a non-harmonic function can be chosen.

**Lemma A.1** (Existence of Base Function). *Let $\mathcal{M}$ be a non-degenerate differential operator on $\Omega \times [0, T]$, where $\Omega \subset \mathbb{R}^n$ is a domain. There exists a function $v \in C^\infty(\Omega \times [0, T])$ such that:*

$$\mathcal{M}(v) \neq 0 \quad \textit{for almost every } (x, t) \in \Omega \times [0, T].$$

*Proof.* Since $\mathcal{M}$ is non-degenerate (i.e., not identically zero), there exists at least one function $w \in C^\infty(\Omega \times [0, T])$ and a point $(x_0, t_0) \in \Omega \times [0, T]$ such that:

$$\mathcal{M}(w)(x_0, t_0) \neq 0.$$

By continuity of $\mathcal{M}(w)$ (assuming smooth coefficients for $\mathcal{M}$), there is an open neighborhood $U \subset \Omega \times [0, T]$ around $(x_0, t_0)$ where $\mathcal{M}(w) \neq 0$.

Construct a smooth bump function $\phi \in C^\infty(\Omega \times [0, T])$ with: $\phi \equiv 1$ on a smaller neighborhood $V \subset U$, and $\phi \equiv 0$ outside $U$. Define $v_0 = \phi \cdot w$. Then $\mathcal{M}(v_0) = \mathcal{M}(\phi w)$ is non-zero on $V$ and smooth everywhere. Let $\{(x_k, t_k)\}_{k=1}^\infty$ be a countable dense subset of $\Omega \times [0, T]$. For each $k$, repeat the above construction to obtain a function $v_k \in C^\infty(\Omega \times [0, T])$ such that: $\mathcal{M}(v_k) \neq 0$ in a neighborhood $U_k$ of $(x_k, t_k)$, $\text{supp}(v_k) \subset U_k$, and the supports $\{U_k\}$ are pairwise disjoint.

Define the function:

$$v = \sum_{k=1}^\infty \epsilon_k v_k,$$

where $\epsilon_k > 0$ are chosen such that the series converges in $C^\infty(\Omega \times [0, T])$ (e.g., $\epsilon_k = 2^{-k} / \max\{\|v_k\|_{C^k}, 1\}$).

The set $\bigcup_{k=1}^\infty U_k$ is open and dense in $\Omega \times [0, T]$. Since $\mathcal{M}(v) \neq 0$ on this dense open set, the zero set $Z = \{(x, t) : \mathcal{M}(v)(x, t) = 0\}$ is contained in the complement of $\bigcup_{k=1}^\infty U_k$, which is nowhere dense and hence has Lebesgue measure zero. Therefore:

$$\mathcal{M}(v) \neq 0 \quad \text{for almost every } (x, t) \in \Omega \times [0, T].$$

$\square$

**Lemma A.2** (Local Correction Functions). *Let $\mathcal{M}$ be a non-degenerate differential operator on $\Omega \times [0, T]$, and let $\chi^* = \{(x_1^*, t_1^*), \ldots, (x_N^*, t_N^*)\} \subset \Omega \times [0, T]$. There exist smooth functions $\{w_i\}_{i=1}^N \subset C^\infty(\Omega \times [0, T])$ and radii $\epsilon_1, \ldots, \epsilon_N > 0$ such that for each $i$:*

*1. Compact Support: $\text{supp}(w_i) \subset B_{\epsilon_i}(x_i^*, t_i^*)$,*

*2. Non-Vanishing Action: $\mathcal{M}(w_i)(x_i^*, t_i^*) \neq 0$,*

*3. Disjoint Supports: $B_{\epsilon_i}(x_i^*, t_i^*) \cap B_{\epsilon_j}(x_j^*, t_j^*) = \emptyset$ for $i \neq j$.*

*Proof.* Let $d_{\min} = \min_{i \neq j} \text{dist}\big((x_i^*, t_i^*), (x_j^*, t_j^*)\big)$ be the minimal distance between distinct points in $\chi^*$. For all $i$, choose radii $\epsilon_i > 0$ such that:

$$\epsilon_i < \frac{d_{\min}}{2}.$$

This ensures the balls $B_{\epsilon_i}(x_i^*, t_i^*)$ are pairwise disjoint.

For each $(x_i^*, t_i^*)$, since $\mathcal{M}$ is non-degenerate, there exists a smooth function $f_i \in C^\infty(\Omega \times [0, T])$ such that:

$$\mathcal{M}(f_i)(x_i^*, t_i^*) \neq 0.$$

This is because, when $\mathcal{M}$ is non-degenerate, its action cannot vanish on all smooth functions at $(x_i^*, t_i^*)$. For instance, if $\mathcal{M}$ contains a derivative $\partial_{x_k}$, take $f_i = x_k$ near $(x_i^*, t_i^*)$.

Then for each $i$, construct a smooth bump function $\phi_i \in C^\infty(\Omega \times [0, T])$ satisfying:

1. $\phi_i \equiv 1$ on $B_{\epsilon_i/2}(x_i^*, t_i^*)$,

2. $\phi_i \equiv 0$ outside $B_{\epsilon_i}(x_i^*, t_i^*)$,

3. $0 \leq \phi_i \leq 1$ everywhere.

Therefore, define the localized function:

$$w_i = \phi_i \cdot f_i.$$

By construction:

1. $\text{supp}(w_i) \subset B_{\epsilon_i}(x_i^*, t_i^*)$,

2. $w_i = f_i$ on $B_{\epsilon_i/2}(x_i^*, t_i^*)$, so

$$\mathcal{M}(w_i)(x_i^*, t_i^*) = \mathcal{M}(f_i)(x_i^*, t_i^*) \neq 0.$$

Since $\epsilon_i < \frac{d_{\min}}{2}$, the distance between any two balls $B_{\epsilon_i}(x_i^*, t_i^*)$ and $B_{\epsilon_j}(x_j^*, t_j^*)$ is at least $d_{\min} - 2\epsilon_i > 0$. Thus, the supports of $w_i$ and $w_j$ are disjoint for $i \neq j$.

Therefore, the functions $\{w_i\}_{i=1}^N$ satisfy all required conditions.

$\square$

We now state the one-dimensional case of Theorem 4.1 here:

**Lemma A.3** (One-Dimensional Case of Theorem 4.1). *Let* $\chi^* = \{(x_1^*, t_1^*), \ldots, (x_N^*, t_N^*)\} \subset \Omega \times [0, T]$. *Then for differential operator* $\mathcal{M}$ *there exist infinitely many functions* $u_\theta : \Omega \to \mathbb{R}$ *parametrized by* $\theta$, *s.t.*

$$\mathcal{M}(u_\theta(x_i^*, t_i^*)) = 0 \quad \text{for } i = 1, \ldots, N,$$

$$\mathcal{M}(u_\theta(x, t)) \neq 0 \quad \text{for a.e. } x \in \Omega \times [0, T] \setminus \chi^*.$$

*Proof.* Define the corrected function:

$$u_\theta = v + \sum_{i=1}^N \alpha_i w_i,$$

where $w_i$ is the local correction function defined in Lemma A.2, $\alpha_i \in \mathbb{R}$ are scalars chosen such that:

$$\mathcal{M}(u_\theta)(x_i^*, t_i^*) = \mathcal{M}(v)(x_i^*, t_i^*) + \alpha_i \mathcal{M}(w_i)(x_i^*, t_i^*) = 0.$$

Since $\mathcal{M}(w_i)(x_i^*, t_i^*) \neq 0$, we can solve for $\alpha_i$:

$$\alpha_i = -\frac{\mathcal{M}(v)(x_i^*, t_i^*)}{\mathcal{M}(w_i)(x_i^*, t_i^*)}.$$

Outside the union of supports $\bigcup_{i=1}^N B_{\epsilon_i}(x_i^*, t_i^*)$, we have:

$$\mathcal{M}(u_\theta) = \mathcal{M}(v) + \sum_{i=1}^N \alpha_i \mathcal{M}(w_i) = \mathcal{M}(v),$$

since $w_i \equiv 0$ outside $B_{\epsilon_i}(x_i^*, t_i^*)$. By construction, $\mathcal{M}(v) \neq 0$ almost everywhere.

The parameters $\theta = (\epsilon_1, \ldots, \epsilon_N, \alpha_1, \ldots, \alpha_N)$ can be varied infinitely by varying $w_i$: The bump functions $w_i$ can be scaled, translated, or reshaped (e.g., Gaussian vs. polynomial) while retaining the properties of Local Correction in Lemma A.2 and varying $\epsilon_i$: For each $i$, choose $\epsilon_i$ from a continuum $(0, \delta_i)$, where $\delta_i$ ensures disjointness.

Thus, the family $\{u_\theta\}$ is uncountably infinite.

The set $\chi^*$ by definition has Lebesgue measure zero in $\Omega \times [0, T]$. The corrections $\sum_{i=1}^N \alpha_i w_i$ are confined to the measure-zero set $\bigcup_{i=1}^N B_{\epsilon_i}(x_i^*, t_i^*)$. Therefore:

$$\mathcal{M}(u_\theta) \neq 0 \quad \text{for a.e. } (x, t) \in \Omega \times [0, T] \setminus \chi^*.$$

$\square$

We now generalize Lemma A.3 to $m$-dimension, to get Theorem 4.1.

**Theorem A.4** (Theorem 4.1). *Let* $\chi^* = \{(x_1^*, t_1^*), \ldots, (x_N^*, t_N^*)\} \subset \Omega \times [0, T]$. *Then for differential operator* $\mathcal{M}$ *there exist infinitely many functions* $u_\theta : \Omega \to \mathbb{R}^m$ *parametrized by* $\theta$, *s.t.*

$$\mathcal{M}(u_\theta(x_i^*, t_i^*)) = 0 \quad for\ i = 1, \ldots, N,$$

$$\mathcal{M}(u_\theta(x, t)) \neq 0 \quad for\ a.e.\ x \in \Omega \times [0, T] \setminus \chi^*.$$

*Proof.* It is trivial to generalize the Lemma A.3 to the case $u_\theta : \Omega \to \mathbb{R}^m$, by constructing:

$$u_\theta = v + \sum_{i=1}^{N} \sum_{j=1}^{m} \alpha_{i,j} w_{i,j},$$

where $\alpha = (\alpha_{i,j}) \in \mathbb{R}^{N \cdot m}$. Adjust $\alpha_{i,j}$ such that:

$$\mathcal{M}(u_\theta)(x_i^*, t_i^*) = \mathcal{M}(v)(x_i^*, t_i^*) + \sum_{j=1}^{m} \alpha_{i,j} \mathcal{M}(w_{i,j})(x_i^*, t_i^*) = 0.$$

This gives a linear system for $\alpha$, which is solvable because the $w_{i,j}$ are linearly independent. $\square$

## B. Linear Time-Varying System

To adjust the given Linear Time-Invariant system to a Linear Time-Varying system, we replace the constant matrices $\bar{A}$, $\bar{B}$, and $C$ with their time-varying counterparts $\bar{A}(k)$, $\bar{B}(k)$, and $C(k)$. The state transition matrix $\bar{A}^{k-i}$ in the LTI system becomes the product of time-varying matrices from time $i$ to $k-1$. The resulting time-varying output equation is:

$$\mathbf{u}_k = C(k)\Phi(k, 0)\mathbf{h}_0 + C(k) \sum_{i=0}^{k} \Phi(k, i)\bar{B}(i)\mathbf{x}_i, \tag{32}$$

where $\Phi(k, i)$ is the state transition matrix from time $i$ to $k$, defined as:

$$\Phi(k, i) = \begin{cases} \bar{A}(k-1)\bar{A}(k-2) \cdots \bar{A}(i) & \text{if } k > i, \\ I & \text{if } k = i. \end{cases} \tag{33}$$

and the term $\Phi(k, 0)\mathbf{h}_0$ represents the free response due to the initial condition $\mathbf{h}_0$.

The summation $\sum_{i=0}^{k} \Phi(k, i)\bar{B}(i)\mathbf{x}_i$ includes contributions from all inputs $\mathbf{x}_i$ up to time $k$, with $\Phi(k, i)\bar{B}(i)$ capturing the time-varying dynamics.

To adjust the Eq. 18 to a Time-Varying system The state transition term $\bar{A}^{k-i}$ becomes the time-ordered product $\Phi(k, i)$, and the output $\mathbf{u}_k$ now explicitly depends on time-varying dynamics. The adjusted equation becomes:

$$\sum_{i=1}^{M} \mathcal{L}_{\mathcal{F}}(u(x_i, k\Delta t)) = \frac{1}{M} \|\mathcal{F}(\mathbf{1}_M \cdot \mathbf{u}_k)\|^2 = \frac{1}{M} \left\| \mathcal{F}\left(\mathbf{1}_M \cdot \mathbf{u}_k = C(k)\Phi(k, 0)\mathbf{h}_0 + C(k) \sum_{i=0}^{k} \Phi(k, i)\bar{B}(i)\mathbf{x}_i\right) \right\|^2. \tag{34}$$

This modification ensures consistency with the Time-Varying system's time-dependent parameters while preserving the structure of the original loss function.

# C. PDEs Setups

## C.1. 1-D Convection

The 1-D convection equation, also known as the 1-D advection equation, is a partial differential equation that models the transport of a scalar quantity $u(x, t)$ (such as temperature, concentration, or momentum) due to fluid motion at a constant velocity $c$. It is a fundamental equation in fluid dynamics and transport phenomena. The equation is given by:

$$
\begin{aligned}
\frac{\partial u}{\partial t} + \beta \frac{\partial u}{\partial x} &= 0, \ \forall x \in [0, 2\pi], t \in [0, 1], \\
u(x, 0) &= \sin x, \ \forall x \in [0, 2\pi], \\
u(0, t) &= u(2\pi, t), \ \forall t \in [0, 1],
\end{aligned}
\tag{35}
$$

where $\beta$ is the constant convection (advection) speed. As $\beta$ increases, the equation will be harder for PINN to approximate. It is a well-known equation with failure mode for PINN. We set $\beta = 50$ following common practice (Zhao et al., 2024; Wu et al., 2024).

The 1-D convection equation's analytical solution is given by:

$$
u_{\text{ana}}(x, t) = \sin(x - \beta t).
\tag{36}
$$

## C.2. 1-D Reaction

The 1-D reaction equation is a partial differential equation that models how a chemical species reacts over time and (optionally) varies along a single spatial dimension. The equation is given by:

$$
\begin{aligned}
\frac{\partial u}{\partial t} - \rho u(1 - u) &= 0, \ \forall x \in [0, 2\pi], t \in [0, 1], \\
u(x, 0) &= \exp(-\frac{(x - \pi)^2}{2(\pi/4)^2}), \ \forall x \in [0, 2\pi], \\
u(0, t) &= u(2\pi, t), \ \forall t \in [0, 1],
\end{aligned}
\tag{37}
$$

where $\rho$ is the growth rate coefficient. As $\rho$ increases, the equation will be harder for PINN to approximate. It is a well-known equation with failure mode for PINN. We set $\rho = 5$ following common practice (Zhao et al., 2024; Wu et al., 2024).

The 1-D reaction equation's analytical solution is given by:

$$
u_{\text{ana}} = \frac{\exp(-\frac{(x-\pi)^2}{2(\pi/4)^2}) \exp(\rho t)}{\exp(-\frac{(x-\pi)^2}{2(\pi/4)^2})(\exp(\rho t) - 1) + 1}.
\tag{38}
$$

## C.3. 1-D Wave

The 1-D wave equation is a partial differential equation that describes how a wave propagates through a medium, such as a vibrating string. We consider such an equation given by:

$$
\begin{aligned}
\frac{\partial^2 u}{\partial t^2} - 4 \frac{\partial^2 u}{\partial x^2} &= 0, \ \forall x \in [0, 1], t \in [0, 1], \\
u(x, 0) &= \sin(\pi x) + \frac{1}{2} \sin(\beta \pi x), \ \forall x \in [0, 1], \\
\frac{\partial u(x, 0)}{\partial t} &= 0, \ \forall x \in [0, 1], \\
u(0, t) = u(1, t) &= 0, \ \forall t \in [0, 1],
\end{aligned}
\tag{39}
$$

where $\beta$ is a wave frequency coefficient. We set $\beta$ as 3 following common practice (Zhao et al., 2024; Wu et al., 2024). The wave equation contains second-order derivative terms in the equation and first-order derivative terms in the initial condition, which is considered to be hard to optimize (Wu et al., 2024). This example illustrates that PINNMamba can better capture

the time continuum because its differentiation for time is directly defined by the matrix, whose differential scale is uniform for multiple orders.

The 1-D wave equation's analytical solution is given by:

$$u_{\text{ana}}(x, t) = \sin(\pi x)\cos(2\pi t) + \sin(\beta \pi x)\cos(2\beta \pi t). \tag{40}$$

### C.4. 2-D Navier-Stokes

The 2-D Navier-Stokes equation describes the motion of fluid in two spatial dimensions $x$ and $y$. It is fundamental in fluid dynamics and is used to model incompressible fluid flows. We consider such an equation given by:

$$\frac{\partial u}{\partial t} + \lambda_1(u\frac{\partial u}{\partial x} + v\frac{\partial u}{\partial y}) = -\frac{\partial p}{\partial x} + \lambda_2(\frac{\partial^2 u}{\partial x^2} + \frac{\partial^2 u}{\partial v^2}),$$
$$\frac{\partial v}{\partial t} + \lambda_1(u\frac{\partial v}{\partial x} + v\frac{\partial v}{\partial y}) = -\frac{\partial p}{\partial y} + \lambda_2(\frac{\partial^2 u}{\partial x^2} + \frac{\partial^2 u}{\partial v^2}), \tag{41}$$

where $u(x, y, t)$, $v(x, y, t)$, and $p(x, y, t)$ are the x-coordinate velocity field, y-coordinate velocity field, and pressure field, respectively. We set $\lambda_1 = 1$ and $\lambda_2 = 0.01$ following common practice (Zhao et al., 2024; Raissi et al., 2019).

The 2-dimensional Navier-Stokes equation doesn't have an analytical solution that can be described by existing mathematical symbols, we take Raissi et al. (2019)'s finite-element numerical simulation as ground truth.

### C.5. PINNNacle

PINNacle (Hao et al., 2024) contains 16 hard PDE problems, which can be classified as Burges, Poisson, Heat, Navier-Stokes, Wave, Chaotic, and other High-dimensional problems. We only test PINNmamba on 6 problems, because solving the remaining problems with a sequence-based PINN model will cause an out-of-memory issue, even on the most advanced NVIDIA H100 GPU. Please refer to the original paper of PINNacle (Hao et al., 2024) for the details of the benchmark.

## D. Training Details

**Hyperparameters.** We provide the training hyperparameters of the main experiments in Table 4.

*Table 4.* Hyperparameters for main results.

| Model | Hyperparameter Type | Value |
|---|---|---|
| PINN | network depth | 4 |
| | network width | 512 |
| QRes | network depth | 4 |
| | network width | 256 |
| KAN | network width | [2,5,5,1] |
| | grid size | 5 |
| | grid_epsilon | 1.0 |
| PINNsFormer | # of encoder | 1 |
| | # of decoder | 1 |
| | embedding size | 32 |
| | attention head | 2 |
| | MLP hidden width | 512 |
| | sequence length $k$ | 5 |
| | sequence interval $\Delta t$ | 1e-4 |
| PINNMamba | # of encoder | 1 |
| | embedding size | 32 |
| | $\Delta, B, C$ width | 8 |
| | MLP hidden width | 512 |
| | sequence length $k$ | 7 |
| | sequence interval $\Delta t$ | 1e-2 |

**Computation Overhead.** We report the training time and memory consumption of baseline models and PINNMamba on the convection equation in Table 5.

Table 5. Memory overhead and training time on H100 GPU for solving convection equation.

| Method | Memory Overhead | Training Time |
|---|---|---|
| PINN | 1605 MB | 0.28 s/it |
| QRes | 1561 MB | 0.38 s/it |
| PINNsFormer | 8703 MB | 1.82 s/it |
| KAN | 1095 MB | 2.73 s/it |
| PINNMamba | 7899 MB | 1.99 s/it |

## E. Sensitivity Analysis

PINNMamba can be further improved by hyper-parameters tuning, we test the sub-sequence length, interval and activation selection in this section.

**Sub-sequence Length.** We test the effect of different sub-sequence lengths on model performance. As shown in Table 6, we test the length of 3, 5, 7, 9, 21. Length $k = 7$ achieves the best performance on reaction and wave equations, while $k = 5$ achieves the best performance on convection equation.

Table 6. Results with different Sub-Sequence Length of PINNmamba.

| Length | Convection | | Reaction | | Wave | |
|---|---|---|---|---|---|---|
| | rMAE | rRMSE | rMAE | rRMSE | rMAE | rRMSE |
| 3 | 0.6698 | 0.7271 | 0.0150 | 0.0331 | 0.5288 | 0.533926 |
| 5 | **0.0092** | **0.0099** | 0.0131 | 0.0286 | 0.0278 | 0.0303 |
| 7 | 0.0188 | 0.0201 | **0.0094** | **0.0217** | **0.0197** | **0.0199** |
| 9 | 0.0410 | 0.0444 | 0.0105 | 0.0246 | 0.0343 | 0.0374 |
| 21 | 1.0263 | 1.0596 | 0.0884 | 0.1588 | 0.0458 | 0.0493 |

**Sub-Sequence Interval.** We test the effect of different sub-sequence intervals on model performance. As shown in Table 7, we test the intervals of $2e-3, 5e-3, 1e-2, 1e-1$. The interval $\Delta t = 1e-2$ achieves the best performance on convection and wave equations, while $\Delta t = 5e-3$ achieves the best performance on reaction. Note that, when $\Delta t = 1e-1$, we cannot build the sub-sequence contrastive alignment.

Table 7. Results with different Sub-Sequence Interval of PINNmamba, $k$ is set to 7.

| Interval | Convection | | Reaction | | Wave | |
|---|---|---|---|---|---|---|
| | rMAE | rRMSE | rMAE | rRMSE | rMAE | rRMSE |
| 2e-3 | 0.0249 | 0.0257 | 0.0739 | 0.1389 | 0.1693 | 0.1903 |
| 5e-3 | 0.0243 | 0.0287 | **0.0083** | **0.0185** | 0.2492 | 0.2690 |
| 1e-2 | **0.0188** | **0.0201** | 0.0094 | 0.0217 | **0.0197** | **0.0199** |
| 1e-1 | 1.2169 | 1.3480 | 0.4324 | 0.5034 | 0.0666 | 0.0703 |

**Activation Function.** We test the activation function's effect on the performance of PINNMamba. We report the results of ReLU (Nair & Hinton, 2010), Tanh (Fan, 2000), and Wavelet (Zhao et al., 2024) in Table 8.

Table 8. Results with different activation function in PINNmamba.

| Activation | Convection | | Reaction | | Wave | |
|---|---|---|---|---|---|---|
| | rMAE | rRMSE | rMAE | rRMSE | rMAE | rRMSE |
| ReLU | 0.4695 | 0.4722 | 0.0865 | 0.1583 | 0.4139 | 0.4203 |
| Tanh | 0.4531 | 0.4601 | 0.0299 | 0.0568 | 0.3515 | 0.3539 |
| Wavelet | 0.0188 | 0.0201 | 0.0094 | 0.0217 | 0.0197 | 0.0199 |

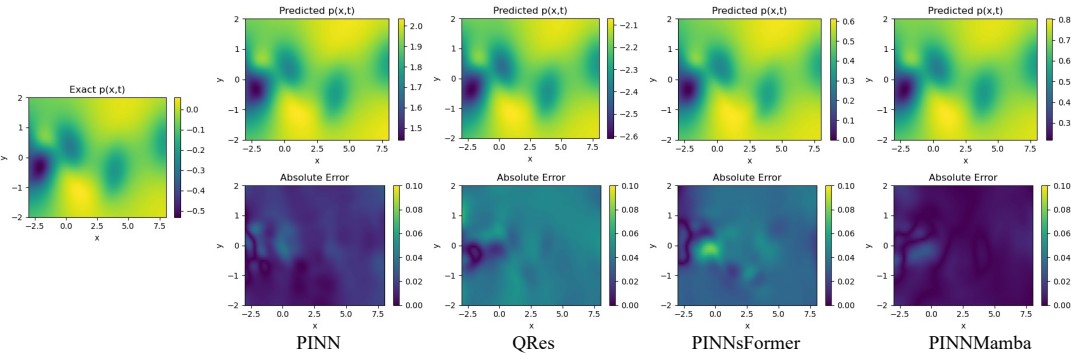

*Figure 11.* The ground truth solution, prediction (top), and absolute error (bottom) on Navier-Stokes equations.

## F. Complex Problem Results

### F.1. 2D Navier-Stokes Equations

Although PINN can already handle Navier-Stokes equations well, we still tested the performance of PINN Mamba on Navier-Stokes equations to check the generalization performance of our method on high-dimensional problems. As shown in Fig. 11, our method achieves good results on Navier-Stokes pressure prediction. Since there is no initial condition information for the N-S equation for pressure, we took the data from the only collocation point for pattern alignment.

### F.2. PINNacle Benchmark

Like PINNsFormer, PINNMamba is a sequence model. The sequence model suffers from Out-of-Memory problems when dealing with some of the problems in the PINNacle Benchmark (Hao et al., 2024), even when running on the advanced Nvidia H100 GPU. We report here the results of the sub-problems for which results can be obtained in Table 9. PINNMamba can solve the Out-of-Memory problem by distributed training over multiple cards, which we leave as a follow-up work.

*Table 9.* Results on PINNacle. Baseline results are from RoPINN paper (Wu et al., 2024). OOM means Out-of-Memroy.

| Equation | PINN | | PINNsFormer | | PINNMamba | |
|---|---|---|---|---|---|---|
| | rMAE | rRMSE | rMAE | rRMSE | rMAE | rRMSE |
| Burgers 1d-C | 1.1e-2 | 3.3e-2 | 9.3e-3 | 1.4e-2 | 3.7e-3 | 1.1e-3 |
| Burgers 2d-C | 4.5e-1 | 5.2e-1 | OOM | OOM | OOM | OOM |
| Poisson 2d-C | 7.5e-1 | 6.8e-1 | 7.2e-1 | 6.6e-1 | 6.2 e-1 | 5.7e-1 |
| Poisson 2d-CG | 5.4e-1 | 6.6e-1 | 5.4e-1 | 6.3e-1 | 1.2e-1 | 1.4e-1 |
| Poisson 3d-CG | 4.2e-1 | 5.0e-1 | OOM | OOM | OOM | OOM |
| Poisson 2d-MS | 7.8e-1 | 6.4e-1 | 1.3e+0 | 1.1e+0 | 7.2e-1 | 6.0e-1 |
| Heat 2d-VC | 1.2e+0 | 9.8e-1 | OOM | OOM | OOM | OOM |
| Heat 2d-MS | 4.7e-2 | 6.9e-2 | OOM | OOM | OOM | OOM |
| Heat 2d-CG | 2.7e-2 | 2.3e-2 | OOM | OOM | OOM | OOM |
| NS 2d-C | 6.1e-2 | 5.1e-2 | OOM | OOM | OOM | OOM |
| NS 2d-CG | 1.8e-1 | 1.1e-1 | 1.0e-1 | 7.0e-2 | 1.1e-2 | 7.8e-3 |
| Wave 1d-C | 5.5e-1 | 5.5e-1 | 5.0e-1 | 5.1e-1 | 1.0e-1 | 1.0e-1 |
| Wave 2d-CG | 2.3e+0 | 1.6e+0 | OOM | OOM | OOM | OOM |
| Chaotic GS | 2.1e-2 | 9.4e-2 | OOM | OOM | OOM | OOM |
| High-dim PNd | 1.2e-3 | 1.1e-3 | OOM | OOM | OOM | OOM |
| High-dim HNd | 1.2e-2 | 5.3e-3 | OOM | OOM | OOM | OOM |

