# OpenReview forum: "Sub-Sequential Physics-Informed Learning with State Space Model"
_ICML.cc/2025/Conference — ICML 2025 poster_

### Official Review · Reviewer_PWyp · 2025-03-11

**Overall Recommendation:** 2

**Summary:**

This paper addresses the two fundamental challenges in training PINN, continuous-discrete mismatch and simplicity bias. The proposed PINNMamba employs 1) State Space Model (SSM) to effectively capture continuous information in discrete temporal sequences and 2) sub-sequence contrastive alignment not to make the model trapping in the over-smooth local optimum (simple but incorrect solution).

**Claims And Evidence:**

I have concerns on the novelty of problem statement. While I agree that the continuous-discrete mismatch and simplicity bias are important to enhance the learnability of PINN, those two challenges are not considered a novel problem. As we know, there has been numerous literatures to model continuous time series from discrete one, shifting the focus of spatio-temporal modeling methods from conventional RNNs to neural ODEs and PDEs. Ultimately, they try to model continuous time series from discrete sampling. Also, I cannot directly connect if this would block the propagation of the “initial condition” (L57), as there could be more causes blocking its propagation. Simplicity bias is also an empirically very well-known problem, when comparing model and data complexity.  Hence, the author’s main problem statement “How can we effectively introduce sequentiality to PINNs?” with a toy example in Figure 1 does not sound a challenging problem.

**Essential References Not Discussed:**

N/A

**Experimental Designs Or Analyses:**

More detailed experiments are required to validate the proposed methods are effective to address the continuous-discrete mismatch and simplicity bias.  For example, in section 6.4. Ablation Study, PINNMamba without Sub-Sequence Align (in Table 3) demonstrates the higher loss than the original PINNMamba. This number does not directly indicate the simplicity bias is relieved, and related discussion is missing in the section. At the same time, I think "relieved continuous-discrete mismatch and simplicity bias" are challening to generally validate with few empirical results, and hence it would be nice to present relevant analytical studies.

**Methods And Evaluation Criteria:**

While the motivation to deploy SSM into PINN model makes sense, its technical novelty is limited. It simply adds a SSM module between MLP modules, which could be considered a simple variation from common PINN designs. Sub-sequence contrastive alignment is straightforward and also makes sense. However, again, I doubt if we can say if it significantly differs from well-known temporal contrastive learning. Learning from sub sequences is widely used to efficiently learn long sequences with better stability. Overall, I understand the author’s motivation to use SSM and sub-sequence contrastive alignment, yet find their limited technical novelties.

**Other Comments Or Suggestions:**

N/A

**Other Strengths And Weaknesses:**

N/A

**Questions For Authors:**

N/A

**Relation To Broader Scientific Literature:**

Physics-informed neural networks (PINN) have a wide range of application, as PDEs are the fundamental component of scientific researches and applications. This paper focuses on enhancing the learnability of PINN, particularly when the discrete training data degrades its training process and the model easily traps in a naive local minimum. Such scenarios can often happen in practical setups.

**Theoretical Claims:**

I appreciate the author formulate Theorem 3.1. and its proof in Appendix A. However, it is not clearly connected to the proposed claim (i.e., continuous-discrete mismatch). There could be of course more solutions that satisfy discrete samples, as they are partial observations from original continuous source. However, I think it does not directly indicate that continuous-discrete mismatch is the key challenge in propagating initial condition in training PINN.

---

> ### Author Rebuttal · Authors · 2025-03-31
>
> We sincerely appreciate the constructive comments from reviewer PWyp and the time spent on reviewing this paper. We address the questions and clarify the issues accordingly as described below.
>
> >**[C&E:]** I have concerns on the novelty .... with a toy example in Figure 1 does not sound a challenging problem.
>
> **[Re:]** To understand our work's contribution, one must first understand the current status of research on **PINN failure modes**, which is the core problem we are addressing. PINN failure mode is not as simple as a regular learnability problem. It is a problem unique to PINN and was first identified by Krishnapriyan et al in 2021 NeurIPS. PINN failure modes refer to a phenomenon where the residual loss of a PINN is extremely low, but its average error (even on training collection points) is extremely high. This phenomenon is distinguished from concepts like local minimal (PINN failure modes loss can be as small as ground truth), overfitting, and gradient vanishing, in which the core problem is optimization or generalization. The essential difficulty of this problem is that ground truth is unknowable, and the only known fully optimized PDE-based governing loss is unreliable. In real-world applications, this can result in serious misestimation of the system.
>
> The central contribution of our paper is a unique understanding of this unique phenomenon. The fact that the continuous-discrete mismatch and simplicity bias are well known does not mean that their explanatory role for PINN failure modes is well established. The ideas in the traditional spatiotemporal modeling do not necessarily apply to PINN, since training a PINN is a data-free and ground-truth-unknown problem. A great deal of discussion has arisen in the research community regarding failure modes for PINNs, but to the best of our knowledge, including PINNsFormer, which introduced the Transformer to PINNs, has failed to intrinsically understand and address the PINN failure modes. The distortions in the propagation of the initial conditions we propose are empirically observable, as shown in Fig.1, 2, 6, 7, and 8. We propose that continuous-discrete mismatch and simplicity bias are the core causes of these propagation failures, which is a completely novel understanding of the subject.
>
> We also need to point out that the convection problem shown in Fig.1 is a well-known challenging failure mode case, proposed by [1] and used by several related works.
>
> >**[M&EC:]** technical novelty is limited.
>
> **[Re:]**: Our contribution lies not only in this simple and effective SSM-based approach but also in the deep understanding of PINN failure modes. Our proposed SSM-based approach is an embodiment of this understanding. Without this understanding, such an approach that can address failure modes in a wide range of equations would never have been produced. In general, we think the machine learning community always appreciates and prefers simple and effective methods.
>
> Sub-sequence contrastive alignment differs from temporal contrastive learning in that, instead of a model learning contrastively for multiple similar data/frame features, it aims to allow different stages of time-varying SSM to form a consensus on the prediction of spatiotemporal collection points. Sub-sequence contrastive alignment emphasizes the ability of the model to inherit information at subsequent times and the ability to form an SMM-step-width agreement, while temporal contrastive learning focuses on representation learning of similar data features, which are not available in PINN.
>
> >**[TC:]**  Theorem 3.1 ... does not directly indicate that continuous-discrete mismatch is the key challenge in propagating initial condition in training PINN.
>
> **[Re:]** What we need to show through Theorem 3.1 is that continuous-discrete mismatch may lead to disconnections in the pattern propagation from the initial condition if only the losses at discrete collection points are optimized. This is because a pattern defined by a point may only act in its small neighborhood, and this neighborhood may not contain any other collection points, which can therefore cause the failure of propagation.
> Also, a concurrent work, ProPINN [2] observed a lower gradient correlation phenomenon, which is empirical evidence for our proposal that continuous-discrete mismatch is the cause of propagation failure.
>
> >**[ED & A:]** Detailed Ablation and Discussion.
>
> **[Re:]** We add some ablation studies to discuss the combinatorial effect. See response to reviewer yZBS. Due to the rebuttal limit, we can't include a more detailed analytical discussion of these ablations but will do so in the next release.
>
> Given these explanations, we sincerely invite you to reconsider the rating of our work.
>
> [1] Krishnapriyan, et al. "Characterizing possible failure modes in physics-informed neural networks." NeurIPS 2021
>
> [2] Wu, et al. "ProPINN: Demystifying Propagation Failures in Physics-Informed Neural Networks." arXiv:2502.00803

---

### Official Review · Reviewer_Ce7f · 2025-03-13

**Overall Recommendation:** 3

**Summary:**

The authors find the reason of failure modes of PINNs is the dismatch between continous nature of PDE and the discrete nature of sampled observations, and the simplicity bias of PINNs fails fixing this gap. To address this, they propose to use Mamba's sequence modeling ability and enhance it with alignment.

### **After rebuttal**

Thanks to the authors for their feedback. I think my concerns have been addressed, and I would like to keep my score 3 unchanged.

**Claims And Evidence:**

Yes. The experimental results demostrate the claims of good performance.

**Essential References Not Discussed:**

Based on my limited knowledge, I am unable to identify any missing essential related works.

**Experimental Designs Or Analyses:**

Yes. The experimental designs and alalyses are sound.

**Methods And Evaluation Criteria:**

Yes.

**Other Comments Or Suggestions:**

1. The theoretical result Theorem 3.1 is interesting. It formalizes the well-known issue that optimizing PDE constraints at discrete points does not ensure global correctness. Though interesting, it is not novel. There is also no theory justifying why PINNMamba outperforms PINN.

2. The proposal cannot fundamentally resolve simplicity bias. While sub-sequence contrastive alignment mitigates the issue by enforcing consistency across overlapping sub-sequences, it does not eliminate the tendency of neural networks to favor simpler solutions.

**Other Strengths And Weaknesses:**

Strengths:

1. Proposes PINNMamba, integrating Selective SSM (Mamba) with PINNs to improve temporal information propagation and mitigate failure modes.
2. Demonstrates significant performance gains on benchmark PDEs, reducing errors compared to previous PINN architectures.
3. This paper is very clearly written.

Weaknesses:
1. Claims to resolve the continuous-discrete mismatch but still relies on a discretized version of Mamba, which remains dependent on time-step selection. Therefore, the continuous-discrete mismatch still persists, making this an ad-hoc rather than a fundamental solution.
2. Although the authors mentioned various methods for addressing failure modes in the related works, including optimization techniques (Wu et al., 2024; Wang et al., 2022a), adaptive sampling strategies (Gao et al., 2023; Wu et al., 2023), model architectures (Zhao et al., 2024; Cho et al., 2024; Nguyen et al., 2024b), and transfer learning approaches (Xu et al., 2023; Cho et al., 2024), they did not provide a thorough comparison with these methods in their analysis or experiments.

**Questions For Authors:**

1. Novelty: How is this work fundamentally different from PINNsFormer (Zhao et al., 2024), given that it seemly replaces Transformers with Mamba, a common modification in recent models?

2. Writing: Is there a typo in Line 326 "$(x,k+\Delta t)$"?

3. In Page 17, the authors say "because solving the remaining problems with a sequence-based PINN model will cause an out-of-memory issue". Does this mean PINNMamba needs more memory than traditional solutions?

**Relation To Broader Scientific Literature:**

The paper is based on related work on PINN failure modes (Krishnapriyan et al., 2021), state-space models (Gu & Dao, 2023), and sequence modeling for PDEs (Zhao et al., 2024). However, it lacks thorough comparisons with adaptive sampling (Wu et al., 2023; Gao et al., 2023) and optimization strategies like RoPINN (Wu et al., 2024) and NTK-based methods (Wang et al., 2022), which also address PINN failure modes.

**Theoretical Claims:**

I checked the proof.

---

> ### Author Rebuttal · Authors · 2025-03-31
>
> We sincerely appreciate the constructive comments from reviewer Ce7f and the time spent on reviewing this paper. We address the questions and clarify the issues accordingly as described below.
>
> >**[W1]**: Claims to resolve the continuous-discrete mismatch but still relies on a discretized version of Mamba..
>
> **[Response to W1]**: We need to clarify that we do not claim to solve the continuous-discrete mismatch. In fact, we think that the computer's finite precision inherently makes it impossible to describe continuous systems (including continuous-time SSM).
>
> Instead, we can only try to mitigate the effect of this mismatch. A discretized SSM is a description of the behavior of a continuous SSM. The point here is that even discrete-time SSM **describes the dynamic in a continuous/differential manner** since there is a set of rules for a direct mapping between a discretized SSM and a continuous SSM. Although it is actually an approximation of continuous dynamics, this ability to directly approximate/describe continuous dynamics is not available with traditional neural network models such as MLP and Transformer.
>
> Indeed, SSM may not be the ultimate solution to this problem, but we have still taken a huge step towards solving continuous-discrete mismatch. We hope that our work will inspire more researchers to achieve a better solution to this problem.
>
> >**[W2]**: Comparsion with other methods.
>
> **[Response to W2]**: Since our approach is an on model architecture, we have mainly listed multiple model architectures as baselines. But we do value your opinion and we will include the following results in the next release. Since the experimental settings of some of the papers differ from ours and the source code is not published, we report those works that we successfully reproduced.
>
> Model|Convection rMAE|Convection rRMSE| Reaction rMAE | Reaction rRMSE | Wave rMAE| Wave rRMSE|
> |-|-|-|-|-|-|-|
> |RoPINN (Wu. 2024 NeurIPS) (Optimization)|0.635| 0.720|0.056 | 0.095|0.063 |0.064|
> |P2INN (Cho. 2024 ICML) (Model Archi)|0.1023| 0.1035|0.0098|0.0227|0.2134|0.2157|
> |DCGD (Hwang. 2024 NeurIPS) (Optimization)|0.0232|0.0246|0.9780|0.9800|OOM|OOM|
> |R3 (Daw. 2023 ICML)(Sampling)|0.0267|0.0277
> |PINNMamba|0.0188 |0.0201|0.0094| 0.0217|0.0197|0.0199|
>
> >**[C&S1]**: Theorem is not novel. There is also no theory justifying why PINNMamba outperforms PINN.
>
> **[Response to C&S1]**: Theorem 3.1 is very intuitive, so we also tried to find a direct proof of Theorem 3.1 when preparing the manuscript, but failed. We would welcome some relevant references from the reviewers.
>
> As for a theory justifying why PINNMamba outperforms PINN. We agree that an in-depth theoretical analysis of the proposed method is helpful to better understand our model. But to be honest, currently, we could not find a good theorem to prove rigorously such mechanisms in deep networks, since theoretically analyzing a complex system like PINNMamba is very difficult. However, we still have some intuitive and empirical analysis to show the methodology and philosophy behind our continuous-discrete mismatch and simplicity bias perspective. We hope such analysis can help readers to better understand our motivation and provide some guidance to design better models in future research.
>
> >**[C&S2]**: The proposal cannot fundamentally resolve simplicity bias.
>
> **[Response to C&S2]**: As previous work[1,2,3] has pointed out, simplicity bias is an intrinsic problem for neural networks. Currently, there is no single way to completely solve this problem, only to evade or mitigate it.
>
> [1] H Shah. The pitfalls.. NeurIPS 2020
>
> [2] D Teney. Evading the simplicity bias.. CVPR 2022
>
> [3]R Tiwari. Overcoming simplicity bias.. ICML 2023
>
> >**[Q1]**: Novelty: How is this work fundamentally different from PINNsFormer, given that it seemly replaces Transformers with Mamba...
>
> **[Response to Q1]**: Our work is not a simple block-wise replacement. Although both works are based on model architecture, they are quite different in the following points.
>
> 1. Macro Architecture: PINNsFormer employs an Encoder-Decoder architecture, which we found unnecessary for PINN. So in PINNMamba, we use an Encoder-Only archi for better performance and efficiency.
>
> 2. Sequence Modeling: As shown in Fig.5, PINNsFormer uses a pseudo-sequence, which is not a real collection points sequence, instead, a regional extrapolation, which cannot propagate information. We build collection point sub-sequences and an alignment to realize info propagation across time.
>
> 3. Understanding: Our work provides a deeper understanding of PINN failure modes that it is caused by C-D mismatch and simplicity bias. We believe it can inspire more valuable work in the future.
>
> >**[Q2]**: typo
>
> **[Response to Q2]**: It should be $(x,t+k\Delta t)$. Will fix.
>
> >**[Q3]**: Memory Usage
>
> **[Response to Q3]**: We greatly optimized memory usage in a follow-up. The OOM is no longer an issue when using a reduced model size. See response to reviewer yZBS.

---

### Official Review · Reviewer_yZBS · 2025-03-14

**Overall Recommendation:** 3

**Summary:**

This paper proposes PINNMamba, enabling PINNs with state-space model's continuous-discrete capability to address the limitations (simplicity bias and continuous-discrete mismatch) of existing PINNs.

**Claims And Evidence:**

1. Mainstream PINNs predominantly use MLPs and suffer from the inability to accurately propagate physical patterns informed by initial conditions. Evidenced by a toy example in Fig. 1.

2. PINNs have issues of continuous-discrete mismatch and simplicity bias. Evidenced by Section 3.

**Essential References Not Discussed:**

N/A

**Experimental Designs Or Analyses:**

The authors evaluated the approach on three public benchmarks (convection, wave, and reaction equations) and compared it with other approaches such as PINN, QRes, PINNsFormer, and Kan.

**Methods And Evaluation Criteria:**

The authors propose to use SSM to address the continuous-discrete mismatch and sub-sequential alignment to improve the simplicity bias limitation.

**Other Comments Or Suggestions:**

See the questions section.

**Other Strengths And Weaknesses:**

Strengths:
1. The paper is well-written and easy to follow.
2. From the results presented by the authors, the proposed approach consistently outperforms other approaches in the three tested benchmarks.

Weaknesses:
1. The proposed approach is computationally expensive, as shown in Table 5 of the supplementary material. The memory overhead is the second largest, and the training time is seven times slower than that of PINN.

**Questions For Authors:**

1. From the sensitivity analysis, the proposed approach is very sensitive to the sub-sequence length, especially comparing the length of 3 and 5 (about 70 times smaller). Are there any explanations for such a big gap?

2. The current ablation study only removes one component at a time, making it difficult to see the effect of other combinations of proposed components.

3. Given that the computational overhead of the proposed PINNMamba encoder is significantly higher than PINN, it would be interesting to see the performance comparison using a similarly sized model to PINN (i.e., with a smaller embedding size).

**Relation To Broader Scientific Literature:**

Addressing the failure modes in PINNs can potentially be applied to a wide range of scientific and engineering disciplines, such as computational fluid dynamics.

**Theoretical Claims:**

Theorem 3.1 is proposed to show the continuous-discrete mismatch failure mode of PINN by demonstrating that there exist infinitely many functions when given a discrete collection of points.

---

> ### Author Rebuttal · Authors · 2025-03-31
>
> We sincerely appreciate the constructive comments from reviewer yZBS and the time spent on reviewing this paper. We address the questions and clarify the issues accordingly as described below.
>
> >**[W1]**: The proposed approach is computationally expensive, as shown in Table 5 of the supplementary material. The memory overhead is the second largest, and the training time is seven times slower than that of PINN.
>
> >**[Q1]**: From the sensitivity analysis, the proposed approach is very sensitive to the sub-sequence length, especially comparing the length of 3 and 5 (about 70 times smaller). Are there any explanations for such a big gap?
>
> **[Re W&Q1]** We answer these two questions together. We start by fixing a bug in Table 6. In our follow-up experiments, we found an error in the experimental data for sequence lengths 3 and 5 in Table 6. The reason for this error is that we accidentally reduced the numerical precision of the calculations when performing these two sets of experiments, which led to an unexpected performance degradation of PINNMamba when using these two sets of parameters. So the conclusion of the original submission about the sensitivity of sequence length needs to be corrected, and we will fix this in the next release. Our latest experimental data are shown in the table below.
>
> Convection:
>
> Length|MLP width| rMAE| rRMSE | Mem. | Time/iter|
> |---|---|---|---|---|---|
> |3|512|0.0102|0.0126|4042MB|1.10s|
> |5|512|0.0059|0.0068|6020MB|1.59s|
> |7|512|0.0188|0.0201|7899MB|1.99s|
>
> Reaction:
>
> Length|MLP width| rMAE| rRMSE | Mem. | Time/iter|
> |---|---|---|---|---|---|
> |3|512|0.0164|0.0352|4042MB|0.90s|
> |5|512|0.0109|0.0244|6020MB|1.25s|
> |7|512|0.0094|0.0217|7899MB|1.56s|
>
> This group of experiments shows that our model can reduce computational and memory overhead by reducing the sequence length. There is a slight accuracy degradation on the reaction problem, but the performance is even better on the convection problem. Nevertheless, they successfully address the failure modes in all cases.
>
> On the other hand, we found that setting the MLP layer's width to 512 is unnecessary. Reducing the MLP width from 512 to 32 yields a model that still solves the failure modes.
>
> Convection:
> Length|MLP width| rMAE| rRMSE | Mem. | Time/iter|
> |---|---|---|---|---|---|
> |3|32|0.0140|0.0167|1900MB|0.79s|
> |5|32|0.0284|0.0321|2586MB|1.16s|
> |7|32|0.0240|0.0269|3310MB|1.35s|
>
> Reaction:
> Length|MLP width| rMAE| rRMSE | Mem. | Time/iter|
> |---|---|---|---|---|---|
> |3|32|0.0042|0.0085|1900MB|0.62s|
> |5|32|0.0032|0.0069|2586MB|0.82s|
> |7|32|0.0060|0.0126|3310MB|1.01s|
>
> If we set sequence length to 3 and MLP width to 32, the memory usage is 1900MB, only slightly larger than 1605MB of vanilla PINN. Given the addressing of failure modes and an rMAE that is only 1/60th of vanilla PINN's, we consider this slight increase in memory and computational overhead to be insignificant. Our model is robust w.r.t sequence length and MLP width. Model with smaller sequence length and MLP width can also eliminate failure modes. This further enhances the generalizability of our approach.
>
> The gap between 3 and 5 sequence length is in fact not large, but when it is 1, it degrades to PINN and failure modes present again. We sincerely apologize for this confusing.
>
> >**[Q2]**: The current ablation study only removes one component at a time, making it difficult to see the effect of other combinations of proposed components.
>
> **[Re Q2]**:
> We added the following experiments.
>
> Model|Convection rMAE|Convection rRMSE| Reaction rMAE | Reaction rRMSE | Wave rMAE| Wave rRMSE|
> |-|-|-|-|-|-|-|
> |PINNMamba|0.0188 |0.0201|0.0094| 0.0217|0.0197|0.0199|
> |-Sub Seq Align & Time Varying SSM|0.1534|0.1572|0.0333|0.0351|0.0701| 0.0702|
> |-Sub Seq Align & SSM|0.9833|0.9836|0.9801|0.9821|0.4211| 0.4431|
> |-Sub Seq Align & Wavelet|0.5000|0.5021|0.0345|0.0399|0.3519| 0.3573|
> |-Time Varying SSM & Wavelet|0.3921|0.3971|0.0287|0.0287|0.2919| 0.2873|
> |-SSM & Wavelet|1.0263|1.0348|0.9987|0.9988|0.5137| 0.5222|
> |-Sub Seq Align & Time Varying SSM & Wavelet|0.3472|0.3521|0.0487|0.0534|0.3437| 0.3492|
> |-Sub Seq Align & SSM & Wavelet|1.2263|1.2748|0.9821|0.9833|0.5421| 0.5453|
>
> These experimental data show that elimination of continuous-discrete mismatch with SSM is the most important factor, illustrating the significance of the PINNMamba model. Also, Sub Sequence Alignment and Time Varying SSM play an important role in eliminating simplicity bias, and the use of wavelet activation function leads to better numerical precision.
>
> >**[Q3]**: ... it would be interesting to see the performance comparison using a similarly sized model to PINN (i.e., with a smaller embedding size).
>
> **[Re Q3]**: For the convection problem, we further reduce the width of MLP and Mamba to 8. This reduces the memory overhead of the model to 1300MB, and the average optimization time per iteration to 0.23s (less than PINN). The rMAE is 0.0262, the rRMSE is 0.0346. It addresses the failure modes and outperforms all baselines.

---

### Official Review · Reviewer_PwV2 · 2025-03-17

**Overall Recommendation:** 3

**Summary:**

The paper introduces PINNMamba, a framework that integrates state space models (SSMs) into physics-informed neural networks (PINNs) to address failure modes in solving partial differential equations (PDEs). It identifies two key issues: the continuous-discrete mismatch, which disrupts initial condition propagation, and simplicity bias, which leads to over-smoothed solutions. PINNMamba mitigates these by using SSMs for continuous-discrete articulation and sub-sequence modeling to enhance pattern propagation. Experimental results show that PINNMamba outperforms existing PINN architectures in solving PDEs with improved accuracy and generalization.

**Claims And Evidence:**

The paper's claims are supported by theoretical analysis, empirical experiments, and comparative evaluations. PINNMamba's ability to mitigate continuous-discrete mismatch and simplicity bias is justified through theory and demonstrated by improved accuracy over baselines. The effectiveness of sub-sequence modeling and state space models (SSMs) is validated through ablation studies and performance metrics.

**Essential References Not Discussed:**

The paper provides a strong review of PINNs but could benefit from discussing other machine learning approaches for solving PDEs, particularly graph neural network (GNN)-based methods like MeshGraphNet. Including a discussion of such methods would provide a more comprehensive view of the broader landscape of neural network-based PDE solvers.

**Experimental Designs Or Analyses:**

The experimental designs and analyses appear sound. The paper evaluates PINNMamba on standard PDE benchmarks, compares it against multiple baselines, and includes ablation studies to validate key design choices. The metrics used, such as relative MAE and RMSE, are appropriate for assessing model accuracy. I did not identify any major issues.

**Methods And Evaluation Criteria:**

The proposed methods and evaluation criteria are appropriate for the problem and align with prior works. The use of state space models (SSMs) and sub-sequence modeling effectively addresses known limitations in PINNs, and the evaluation on standard PDE benchmarks ensures comparability with existing approaches. The inclusion of ablation studies and comparisons with multiple baselines further supports the validity of the methodology.

**Other Comments Or Suggestions:**

- Reorganize Related Works: Move the related works section from the appendix into the main text for better context on contributions.
- Condense Writing in Methods/Intro: The methods and introduction sections could be more concise to improve readability. Some results, such as Table 2, could be moved to the appendix.

**Other Strengths And Weaknesses:**

The authors placed related works in the appendix, which I find unusual and not ideal. Integrating it into the main text would provide better context for their contributions. They could reduce the length of the methods and introduction sections to make space. Additionally, some results, such as Table 2 (which studies the effect of training strategies), could be moved to the appendix to streamline the main presentation.

**Questions For Authors:**

The paper has a major limitation in memory usage, which the authors acknowledge. However, a deeper analysis of the computational bottleneck is needed to study which operations caused the issue. Additionally, would reducing the sequence length or modifying memory-intensive operations help mitigate these issues? Given that PINNsFormer also encounters OOM errors, is there a fundamental limitation in sequence-based PINNs and would this prevent the method to scale up to large scale simulations?

**Relation To Broader Scientific Literature:**

The paper builds on prior work in PINNs and addresses known failure modes by incorporating state space models (SSMs) and sub-sequence modeling. It aligns with existing research on improving PINN stability and accuracy, particularly studies that explore sequential modeling and optimization strategies to mitigate simplicity bias and continuous-discrete mismatch.

**Theoretical Claims:**

I do not specialize in PINNs, so I can only perform a general check of the theoretical claims. The proofs appear logically structured and follow standard mathematical reasoning, but I rely on other reviewers with more expertise in this area to verify their correctness in detail.

---

> ### Author Rebuttal · Authors · 2025-03-31
>
> We sincerely appreciate the constructive comments from reviewer PwV2 and the time spent on reviewing this paper. We address the questions and clarify the issues accordingly as described below.
>
> >**[W1]**: The authors placed related works in the appendix, which I find unusual and not ideal. Integrating it into the main text would provide better context for their contributions.
>
> **[Response to W1]**: We highly agree with your opinion. We will place the related works section in the main text in the next release since another page will be allowed in camera-ready according to ICML author guidance.
>
> >**[Un RW]**: ... Including a discussion of such methods (NN/Graph-based PDE solvers) would provide a more comprehensive view of the broader landscape of neural network-based PDE solvers.
>
> **[Response]**: We will add the following paragraph in the related works section.
>
> **Learning-Based PDE Solvers.** In addition to Physics-Informed Neural Networks (PINNs), other neural-network-based approaches have emerged for solving PDEs, each offering unique advantages. Graph Neural Networks (GNNs) like MeshGraphNet [1] excel at handling irregular domains by treating computational meshes as graphs, making them particularly effective for complex geometries. Neural operators [2], including Fourier Neural Operators (FNOs) [3] and Graph Neural Operators (GNOs) [4], learn mappings between function spaces, enabling generalization across different PDE parameters without retraining. Hybrid approaches combine neural networks with traditional numerical methods, such as neural finite element techniques [5], to enhance solver efficiency. While PINNs uniquely enable data-free solutions through direct physics constraint enforcement, GNNs, and neural operators provide complementary capabilities - GNNs for mesh-based problems and neural operators for parametric systems. These diverse approaches collectively demonstrate the expanding toolkit of neural-network-based PDE solvers across scientific computing applications.
>
>
> [1] Pfaff, Tobias, et al. "Learning mesh-based simulation with graph networks." in ICLR. 2021.
>
> [2] Kovachki, Nikola, et al. "Neural operator: Learning maps between function spaces with applications to pdes." in JMLR. 2023.
>
> [3] Li, Zongyi, et al. "Fourier Neural Operator for Parametric Partial Differential Equations." in ICLR. 2021.
>
> [4] Li,  et al. "Multipole graph neural operator for parametric partial differential equations." in NuerIPS. 2020.
>
> [5] Hennigh, Oliver, et al. "NVIDIA SimNet™: An AI-accelerated multi-physics simulation framework." International conference on computational science. 2021.
>
> >**[W2]**: reduce the length of the methods, introduction and remove some results to the appendix.
>
> **[Response to W2]**: We will streamline these sections in the next release depending on space.
>
> >**[Q]**: About the memory usage.
>
> **[Response to Q]**: The main reason for the large Memory Usage is that, initially, we set the sequence length to 7 and MLP width to 512 in our main experiment, which was made with the consideration of making a fair comparison with the then SOTA model PINNsFormer. High memory usage is one of the main drawbacks of sequence modeling on PINNs, as the gradient information needs to be preserved for every point in the sequence.
>
> However, we find in a follow-up that our model is robust w.r.t sequence length and MLP width. Our model with a smaller sequence length and MLP width can also eliminate failure modes. This further enhances the generalizability of our approach. To scale to more complex problems, we suggest reducing the sequence length or reducing the MLP width.
>
> Our follow-up experiments show that the model is not very sensitive to sequence length and that a sequence length of 3 makes PINNMamba very effective in combating failure modes. (This is contrary to our results in Table 6 because we found that we had incorrectly set the computational precision in the length 3 and 5 setting in our original sensitivity analysis, resulting in severe performance degradation. We will fix this in the next release.) Adjusting the sequence length to 3 leads to a reduction in the memory overhead of the model by about 57%.
>
> In addition, we found that a large MLP width (512) is not necessary. Setting the width to 32 is sufficient to make PINNMamba effective. This saves up to 53% of memory usage.
> The combination of the two
> adjustment can lead to a reduction in memory consumption from 7899MB to 1900MB on the convection problem.
>
> We added the following experiments. It is worth noting that the model successfully eliminates failure modes under all these settings.
>
> When length is set to 1, PINNMamba degrades to PINN and failure modes present again.
>
> Length|MLP width| rMAE| rMRSE | Memory | Time/iter|
> |---|---|---|---|---|---|
> |3|32|0.0140|0.0167|1900MB|0.79s|
> |5|32|0.0284|0.0321|2586MB|1.16s|
> |7|32|0.0240|0.0269|5932MB|1.35s|
> |3|512|0.0102|0.0126|4042MB|1.10s|
> |5|512|0.0059|0.0068|6020MB|1.59s|
> |7|512|0.0188|0.0201|7899MB|1.99s|

---

### Decision · Program_Chairs · 2025-05-01

**Decision:**

Accept (poster)

**Comment:**

This paper proposes PINNMamba, a novel architecture that integrates state space models (SSMs) with physics-informed neural networks (PINNs) to address two major limitations: the continuous-discrete mismatch and simplicity bias in existing PINNs. The approach introduces sub-sequence alignment and time-varying SSMs to improve information propagation and robustness. While the technical novelty is somewhat incremental—drawing on known modeling tools like Mamba and contrastive alignment—the conceptual contribution is well-articulated and backed by compelling empirical evidence. The authors offer detailed theoretical and empirical analysis, and their ablation studies and follow-up experiments demonstrate the effectiveness and scalability of PINNMamba under varying settings. Several reviewers raised concerns regarding novelty, computational cost, and fairness of comparisons, but the rebuttal addressed these points with additional results and clarifications. The work contributes a practical and well-motivated method with strong results across multiple PDE benchmarks, and despite some limitations, it merits a weak accept.